# Symmetry Teleportation for Accelerated Optimization

**Bo Zhao**
University of California, San Diego
bozhao@ucsd.edu

**Nima Dehmamy**
IBM Research
nima.dehmamy@ibm.com

**Robin Walters**
Northeastern University
r.walters@northeastern.edu

**Rose Yu**
University of California, San Diego
roseyu@ucsd.edu

## Abstract

Existing gradient-based optimization methods update parameters locally, in a direction that minimizes the loss function. We study a different approach, symmetry teleportation, that allows parameters to travel a large distance on the loss level set, in order to improve the convergence speed in subsequent steps. Teleportation exploits symmetries in the loss landscape of optimization problems. We derive loss-invariant group actions for test functions in optimization and multi-layer neural networks, and prove a necessary condition for teleportation to improve convergence rate. We also show that our algorithm is closely related to second order methods. Experimentally, we show that teleportation improves the convergence speed of gradient descent and AdaGrad for several optimization problems including test functions, multi-layer regressions, and MNIST classification. Our code is available at https://github.com/Rose-STL-Lab/Symmetry-Teleportation.

## 1 Introduction

Consider the optimization problem of finding $\text{argmin}_{\boldsymbol{w}}\mathcal{L}(\boldsymbol{w})$, where $\mathcal{L}$ is a loss function and $\boldsymbol{w}$ are the parameters. While finding global minima of $\mathcal{L}(\boldsymbol{w})$ is hard for non-convex problems, we can use gradient descent (GD) to find local minima. In GD we apply the following update rule at every step:

$$\boldsymbol{w}_{t+1} \leftarrow \boldsymbol{w}_t - \eta\nabla\mathcal{L}, \tag{1}$$

where $\eta$ is the learning rate. Gradient descent is a first-order method that uses only gradient information. It is easy to compute but suffers from slow convergence. Second-order methods such as Newton's method use the second derivative to account for the landscape geometry. These methods enjoy faster convergence, but calculating the second derivative (Hessian) can be computationally expensive over high dimensional spaces (Hazan, 2019).

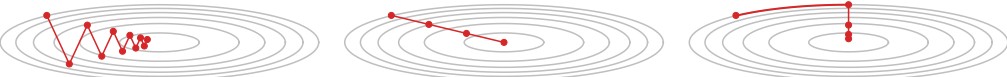

Figure 1: Left to right: gradient descent, second-order methods, proposed method.

In this paper, we propose a new optimization method based on parameter space symmetries, which are group actions on the parameter space that leave the loss unchanged. Our algorithm takes advantage of higher-order landscape geometry but uses only gradient information in most steps, thus avoiding the computational cost of second-order methods.

36th Conference on Neural Information Processing Systems (NeurIPS 2022).

Specifically, we look beyond local optimization and ask: *what if we allow the parameters to make a big jump once in a while?* As shown in Figure 1, during optimization, we teleport the parameters to another point with steeper gradients using a loss-invariant symmetry transformation. After teleportation, the loss stays the same, but the gradient and hence the rate of loss decay changes. The increased magnitude of the gradient can reduce the number of steps required to converge, leading to acceleration of the gradient descent.

The locality of gradient descent is reflected in its formulation in terms of a proximal term; our method circumvents this locality by teleporting to new locations with the same loss. A step of GD is equivalent to the following proximal mapping (Combettes and Pesquet, 2011). Let $\langle \cdot, \cdot \rangle$ denote the inner product, we have

$$\boldsymbol{w}_{t+1} = \operatorname{argmin}_{\boldsymbol{w}} \left\{ \eta \langle (\nabla \mathcal{L})|_{\boldsymbol{w}_t}, \boldsymbol{w} \rangle + \frac{1}{2} \|\boldsymbol{w} - \boldsymbol{w}_t\|_2^2 \right\}. \tag{2}$$

The term $\frac{1}{2}\|\boldsymbol{w} - \boldsymbol{w}_t\|^2$ is the proximal term that keeps $\boldsymbol{w}_{t+1}$ close to $\boldsymbol{w}_t$. Adaptive gradient methods define the proximal term using the Mahalanobis distance $\|\boldsymbol{w} - \boldsymbol{w}_t\|_{G^{-1/2}}^2$ to account for landscape geometry. For example, in AdaGrad, $G$ is the sum of the outer product of gradients (Duchi et al., 2011). Our proposed teleportation technique relaxes the requirement from the proximal term. We teleport to points on the same level set of $\mathcal{L}(\boldsymbol{w}_t)$, but allow $\boldsymbol{w}$ to be far from $\boldsymbol{w}_t$ in Euclidean distance.

In summary, our main contributions are:

- We propose *symmetry teleportation*, an accelerated gradient-based optimization algorithm that exploits symmetries in the loss landscape.
- We derive loss-invariant group actions of test functions and multi-layer neural networks.
- We provide necessary conditions and examples of when teleportation improves convergence rate of the current and subsequent steps.
- We show that our algorithm is closely related to second-order methods.
- Experimentally, we show that teleportation improves the convergence speed of gradient descent and AdaGrad in various optimization problems.

## 2 Related Work

Continuous parameter space symmetries have been identified in neural networks with homogeneous (Badrinarayanan et al., 2015; Du et al., 2018) and radial activation functions (Ganev et al., 2021). The effect of symmetry transformations on gradients has been examined for translation, scale, and rescale symmetries in Kunin et al. (2021). We contribute to this line of work by deriving loss-invariant group actions in multi-layer neural networks with invertible activation functions.

Several works exploit symmetry to facilitate optimization. For example, Path-SGD (Neyshabur et al., 2015) improves optimization in ReLU networks by path regularization. $\mathcal{G}$-SGD (Meng et al., 2019) performs weight updates in a scale-invariant space, which also exploits the symmetry in ReLU networks. Van Laarhoven (2017) analyzes the effect of L2 regularization in batch normalization which gives scale-invariant functions. Bamler and Mandt (2018) transforms parameters in the symmetry orbits to address slow movement along directions of weakly broken symmetry and find the optimal $g \in G$ that minimizes $\mathcal{L}(g \cdot \boldsymbol{w})$. In comparison, we search within $G$-orbits for points which maximize $|d\mathcal{L}(g \cdot \boldsymbol{w})/dt| = \|\nabla \mathcal{L}(g \cdot \boldsymbol{w})\|^2$.

Natural gradient (Amari, 1998), adaptive gradient methods (Duchi et al., 2011; Kingma and Ba, 2015), and their approximations (Martens and Grosse, 2018; Gupta et al., 2018) improve the direction of parameter updates instead of directly transforming parameters. If the group acts transitively on the level set, and we teleport to the point with maximum gradient, then our update direction is the same as that in Newton's method. We prove that our algorithm is connected to second-order optimization methods and show that teleportation can be used to improve these methods empirically.

The concept of neural teleportation was first explored using quiver representation theory (Armenta and Jodoin, 2021; Armenta et al., 2020). These works provide a way to explore level curves of the loss of neural networks and show that random teleportation speeds up gradient descent experimentally and theoretically. Our algorithm improves neural teleportation by searching for teleportation destinations that lead to the largest improvement in the magnitude of gradient.

Several works study how parameter initialization affects convergence rate (Saxe et al., 2014; Tarmoun et al., 2021; Min et al., 2021). If we apply a group transformation only at initialization, our method is similar to that of Tarmoun et al. (2021). We do not guarantee that the transformed parameters lead to faster convergence rate throughout the entire training. However, we accelerate convergence at least for a short time after initialization. Additionally, our method is not restricted to initialization and can be applied at any time during training.

Most contemporary neural networks are overparametrized. While this has been shown to improve generalization (Belkin et al., 2019), an important question is how overparametrization affects optimization. A series of works starting from Arora et al. (2018) show that overparametrization resulting from the depth of a neural network accelerates convergence. Another view is that the symmetry created by overparametrization poses constraints on trajectories in the form of conserved quantities (Głuch and Urbanke, 2021). Additionally, the symmetry generates additional trajectories. When the trajectories created by overparametrization are equivalent, model compression by removing symmetry reduces training time (Ganev et al., 2021). However, when trajectories are not equivalent, gradient flows on some paths are faster than others. We search for the better paths created by overparametrization.

## 3    Symmetry Teleportation

We propose *symmetry teleportation*, an accelerated gradient-based optimization algorithm that exploits symmetries in the loss landscape. Symmetry teleportation searches for the best gradient descent trajectory by teleporting parameters to a point with larger gradients using a group action. The resulting algorithm requires only gradient computations but is able to account for the global landscape geometry, leading to faster loss decay.

Let the group $G$ be a set of symmetries which leave the loss function $\mathcal{L}$ invariant: $\mathcal{L}(g \cdot (\boldsymbol{w}, X)) = \mathcal{L}(\boldsymbol{w}, X)$, where $g \in G$. We perform gradient descent for $t_{max}$ steps. We define an index set $K \subseteq \{0, 1, 2, ..., t_{max} - 1\}$ as a teleportation schedule. At epochs that are in the schedule, we transform parameters using group element $g \in G$ to the location where the gradient is largest, then continue with gradient descent. Algorithm 1 describes the details of this procedure. Note that the loss does not change after teleportation (Line 2-5) since $\mathcal{L}$ is $G$-invariant.

---

**Algorithm 1:** Symmetry Teleportation

**Input:** Loss function $\mathcal{L}(\boldsymbol{w})$, learning rate $\eta$, number of epochs $t_{max}$, initialized parameters $\boldsymbol{w}_0$, symmetry group $G$, teleportation schedule $K$.

**Output:** $\boldsymbol{w}_{t_{max}}$.

1 **for** $t \leftarrow 0$ **to** $t_{max} - 1$ **do**
2     **if** $t \in K$ **then**
3         $g \leftarrow \text{argmax}_{g \in G} \|(\nabla \mathcal{L})|_{g \cdot \boldsymbol{w}_t}\|^2$
4         $\boldsymbol{w}_t \leftarrow g \cdot \boldsymbol{w}_t$
5     **end if**
6     $\boldsymbol{w}_{t+1} \leftarrow \boldsymbol{w}_t - \eta(\nabla \mathcal{L})|_{\boldsymbol{w}_t}$
7 **end for**
8 **return** $\boldsymbol{w}_{t_{max}}$

---

Algorithm 1 can be generalized to apply teleportation to stochastic gradient descent (Appendix A.1). We discuss some design choices below and provide detailed analysis in the next two sections.

When the action of $G$ is continuous, teleportation can be implemented by parameterizing and performing gradient ascent on $g$. For example, the $\text{SO}(2)$ group can be parameterized by the rotation angle $\theta$. Small transformations $g \in \text{GL}_d(\mathbb{R})$ (general linear group) can be parameterized as $g \approx I + \varepsilon T$ where $\varepsilon \ll 1$ and $T$ are arbitrary $d \times d$ matrices. For discrete groups, search algorithms or random sampling can be used to find a group element that improves the magnitude of the gradient.

Although $g \cdot (\boldsymbol{w}, X)$ does not change $X$ in the cases we consider in this paper, Algorithm 1 can be extended to allow transformations on both parameters and data. The group actions on data during teleportations can be precomposed and applied to the input data at inference time. The algorithm extension that allows transformation on the input can be found in Appendix A.2.

The teleportation schedule $K$ is a hyperparameter that determines when to perform teleportation. We define $K$ as a set to allow flexible teleportation schedules, such as with non-fixed frequencies or teleporting only at the earlier epochs.

# 4 Symmetry Groups of Certain Optimization Problems

We give a few practical examples to demonstrate how teleportation can be used to accelerate optimization. Specifically, we first consider two test functions which are often used to evaluate optimization algorithms (Back, 1996). We then derive the symmetries of multi-layer neural networks.

## 4.1 Test functions

**Rosenbrock function.** The Rosenbrock function originally introduced by Rosenbrock (1960) has a characteristic global minimum that is inside a long, narrow, parabolicly-shaped flat valley. Finding the valley is easy but reaching the minimum is difficult. On a 2-dimensional space, the Rosenbrock function has the following form:

$$\mathcal{L}_r(x_1, x_2) = 100(x_1^2 - x_2)^2 + (x_1 - 1)^2. \tag{3}$$

**Booth function.** The Booth function (Jamil and Yang, 2013) is also defined on $\mathbb{R}^2$ and has one global minimum at $(1, 3)$ where the function evaluates to $0$:

$$\mathcal{L}_b(x_1, x_2) = (x_1 + 2x_2 - 7)^2 + (2x_1 + x_2 - 5)^2. \tag{4}$$

The following proposition identifies the symmetry of these two test functions.

**Proposition 4.1.** *The Rosenbrock and Booth functions have rotational symmetry. In other words, there exist action maps $a_r, a_b : \mathrm{SO}(2) \times \mathbb{R}^2 \to \mathbb{R}^2$, such that for all $g \in \mathrm{SO}(2)$,*

$$\mathcal{L}_r(x_1, x_2) = \mathcal{L}_r(a_r(g, [x_1, x_2])) \quad \text{and} \quad \mathcal{L}_b(x_1, x_2) = \mathcal{L}_b(a_b(g, [x_1, x_2])).$$

The exact forms of the action maps are deferred to Appendix B.1.2 and B.1.3. During the teleportation step, our goal is to maximize the gradient within a level set of the loss: $\max_{g \in \mathrm{SO}(2)} \left\| \frac{d\mathcal{L}(g \cdot (x_1, x_2))}{dt} \right\|$.

## 4.2 Multi-layer Neural Networks

Next, we consider feed-forward neural networks. Denote the output of the $m$th layer by $h_m \in \mathbb{R}^{d_m \times n}$, where $d_m$ is the hidden dimension and $n$ is the number of samples. Denote the input by $h_0 = X \in \mathbb{R}^{d_0 \times n}$. In terms of the previous layer output $\tilde{h}_m = W_m h_{m-1}$ where $W_m \in \mathbb{R}^{d_m \times d_{m-1}}$ (we absorb biases into $W_m$ by adding an extra row of ones in $\tilde{h}_{m-1}$), the output $h_m$ is

$$h_m = \sigma(\tilde{h}_m) = \sigma\left(W_m h_{m-1}\right). \tag{5}$$

We assume the activation functions $\sigma : \mathbb{R} \to \mathbb{R}$ are bijections. For instance, Leaky-ReLU is bijective. For other activation, such Sigmoid or Tanh, we analytically extend them to bijective functions (e.g. $\tanh(x) + e^{x-N} - e^{-x+N}$ for $N \gg 1$). For linear activations, we have linear symmetries:

**Proposition 4.2.** *A linear network is invariant under all groups $G_m \equiv \mathrm{GL}_{d_m}$ acting as*

$$g \cdot (W_m, W_{m-1}) = (W_m g^{-1}, g W_{m-1}), \qquad g \cdot W_k = W_k, \quad \forall k \notin \{m, m-1\}. \tag{6}$$

Similar, in the nonlinear case, we want to find $g \cdot W_m$ that keep the outputs $h_k$ for $k \neq m-1$ invariant. With nonlinearity, $\tilde{h}_m = W_m \sigma(W_{m-1} h_{m-2})$. This network has a $\mathrm{GL}_{d_{m-1}}$ symmetry given by

$$g \cdot (W_m, h_{m-1}) = (W_m g^{-1}, g h_{m-1}), \quad g \cdot h_m = W_m g^{-1} g h_{m-1} = h_m. \tag{7}$$

Thus, we have the following proposition regarding symmetries of nonlinear networks:

**Proposition 4.3.** *Assume that $h_{m-2}$ is invertible. A multi-layer network with bijective activation $\sigma$ has a $\mathrm{GL}_{d_{m-1}}$ symmetry. For $g_m \in G_m = \mathrm{GL}_{d_{m-1}}(\mathbb{R})$ the following group action keeps $h_p$ with $p \geq m$ invariant*

$$g_m \cdot W_k = \begin{cases} W_m g_m^{-1} & k = m \\ \sigma^{-1}\left(g_m \sigma\left(W_{m-1} h_{m-2}\right)\right) h_{m-2}^{-1} & k = m-1 \\ W_k & k \notin \{m, m-1\} \end{cases} \tag{8}$$

Proofs can be found in Appendix B.2. Note that this group action depends on the input to the network as well as the current weights of all the lower layers. Yet, since the action keeps the output of upper and lower layers invariant, multiple $G_m$ for different $m$ applied at the same time still keep the network output invariant. The proposition assumes that $h_{m-2}$ is square and full-rank. When $n < d_{m-2}$ and $h_{m-2}$ has rank $n$, (8) (with left inverse of $h_{m-2}$) keeps the loss invariant but does not satisfy the identity axiom of a group action.

## 5 Theoretical Analysis

### 5.1 What symmetries help accelerate optimization

We now discuss the conditions that need to be satisfied for teleportation to accelerate GD. For brevity, we denote all trainable parameters (e.g. $\{W_1, \cdots W_p\}$ for the $p$-layer neural network) by a single flattened vector $\boldsymbol{w} \in \mathbb{R}^n$. Consider a symmetry $G$ of the loss function $\mathcal{L}(\boldsymbol{w})$, meaning for all $g \in G$, $\mathcal{L}(\boldsymbol{w}) = \mathcal{L}(g \cdot \boldsymbol{w})$. We quantify how teleportation by $G$ changes the rate of loss decay, given by

$$\frac{d\mathcal{L}(\boldsymbol{w})}{dt} = \left\langle \frac{\partial \mathcal{L}}{\partial \boldsymbol{w}}, \frac{d\boldsymbol{w}}{dt} \right\rangle = -[\nabla\mathcal{L}]^T \eta \nabla\mathcal{L} = -\|\nabla\mathcal{L}\|_\eta^2, \tag{9}$$

where $\eta$ is the matrix of learning rates (which must be positive semi-definite) and $\|v\|_\eta^2 = v^T \eta v$ is the Mahalanobis norm. A constant learning rate means $\eta = I$.

The following proposition provides the condition a symmetry needs to satisfy to accelerate optimization. (Proofs can be found in Appendix C.1.)

**Proposition 5.1.** *Let $\boldsymbol{w}' = g \cdot \boldsymbol{w}$ be a point we teleport to. Let $J = \partial\boldsymbol{w}'/\partial\boldsymbol{w}$ be the Jacobian. Symmetry teleportation using $g$ accelerates the rate of decay in $\mathcal{L}$ if it satisfies*

$$\left\| \left[J^{-1}\right]^T \nabla\mathcal{L}(\boldsymbol{w}) \right\|_\eta^2 > \|\nabla\mathcal{L}(\boldsymbol{w})\|_\eta^2. \tag{10}$$

If the action of the symmetry group $G \subset GL_n$ is linear we have $\boldsymbol{w}' = g \cdot \boldsymbol{w} = g\boldsymbol{w}$ and $J = g$. It follows that if $G$ is a subgroup of the orthogonal group, $d\mathcal{L}/dt$ will be invariant:

**Corollary 5.2.** *Let $O_\eta$ denote the orthogonal group of invariances of the inverse of the learning rate, $\eta^{-1}$, meaning for $g \in O_\eta$, $g^T \eta^{-1} g = \eta^{-1}$. Then*

$$\forall g \in O_\eta, \qquad \frac{d\mathcal{L}(g \cdot \boldsymbol{w})}{dt} = \frac{\mathcal{L}(\boldsymbol{w})}{dt}. \tag{11}$$

In the simple case where the learning rate is a constant, $O_\eta = O(n)$ becomes the classic orthogonal group (e.g. rotations) with $g^T g = I$. In general, when $g$ preserves the norm of the gradient, symmetry teleportation has no effect on the convergence rate.

From Theorem 3.2 in Hazan (2019), assuming that $\mathcal{L}$ is $\beta$-smooth and is bounded by $|\mathcal{L}| \leq M$, the gradient norm in gradient descent converges as $\frac{1}{2\beta} \sum_{t=1}^{T} \|(\nabla\mathcal{L})|_{\boldsymbol{w}_t}\| \leq 2M$. Teleportation increases $(\nabla\mathcal{L})|_{\boldsymbol{w}_t}$, therefore requiring less time to reach convergence.

### 5.2 Improvement of subsequent steps

Since teleportation moves the parameters to a point with a larger gradient, and subsequent GD steps are local, we would expect that teleportation improves the magnitude of the gradient for a few future steps as well. The following results formalize this intuition (proofs in Appendix C.2).

**Assumption 5.3** (Lipschitz Continuity). *The l2 norm of the gradient is Lipschitz continuous with constant $L \in \mathbb{R}^{\geq 0}$, which is*

$$\left| \left\|\frac{\partial\mathcal{L}}{\partial\boldsymbol{w}_1}\right\|_2 - \left\|\frac{\partial\mathcal{L}}{\partial\boldsymbol{w}_2}\right\|_2 \right| \leq L\|\boldsymbol{w}_1 - \boldsymbol{w}_2\|_2, \tag{12}$$

*where $\boldsymbol{w}_1, \boldsymbol{w}_2$ are two points in the parameter space and $L$ is the Lipschitz constant.*

**Proposition 5.4.** *Consider the gradient descent with a $G$-invariant loss $\mathcal{L}(\boldsymbol{w})$ and learning rate $\eta \in \mathbb{R}^+$. Let $\boldsymbol{w}_t$ be the parameter at time $t$ and $\boldsymbol{w}'_t = g \cdot \boldsymbol{w}_t$ the parameter teleported by $g \in G$. Let $\boldsymbol{w}_{t+T}$ and $\boldsymbol{w}'_{t+T}$ be the parameters after $T$ more steps of gradient descent from $\boldsymbol{w}_t$ and $\boldsymbol{w}'_t$ respectively. Under Assumption 5.3, if $\eta L < 1$, and*

$$\frac{\|\partial\mathcal{L}/\partial\boldsymbol{w}'_t\|_2}{\|\partial\mathcal{L}/\partial\boldsymbol{w}_t\|_2} \geq \frac{(1+\eta L)^T}{(1-\eta L)^T}, \tag{13}$$

*then*

$$\left\|\frac{\partial\mathcal{L}}{\partial\boldsymbol{w}'_{t+T}}\right\|_2 \geq \left\|\frac{\partial\mathcal{L}}{\partial\boldsymbol{w}_{t+T}}\right\|_2. \tag{14}$$

The proposition provides a sufficient condition for teleportation to improve $T$ future steps. The condition is met when $L$ is small, $\eta$ is small, or the initial improvement, $\left\|\frac{\partial\mathcal{L}}{\partial\boldsymbol{w}'_t}\right\|_2 / \left\|\frac{\partial\mathcal{L}}{\partial\boldsymbol{w}_t}\right\|_2$, is large.

## 5.3 Convergence analysis for convex quadratic functions

Teleportation improves the magnitude of gradient for the current step. We have also shown that the magnitude of gradient stays large for a few subsequent steps (Proposition 5.4). In this section, we analyze a class of functions where teleporting once guarantees optimality at all future times. We consider a trajectory optimal if for every point on the trajectory, the magnitude of gradient is at a local maximum in the loss level set that contains the point.

Consider a positive definite quadratic form $\mathcal{L}_A(\boldsymbol{w}) = \boldsymbol{w}^T A \boldsymbol{w}$, where $\boldsymbol{w} \in \mathbb{R}^n$ is the parameter and $A \in \mathbb{R}^{n \times n}$ is a positive definite matrix. The gradient of $\mathcal{L}_A$ is $\nabla\mathcal{L}_A = 2A\boldsymbol{w}$, and the Hessian of $\mathcal{L}_A$ is $H = 2A$. Since $A$ is defined to be positive definite, $\mathcal{L}_A$ is convex. The function $\mathcal{L}_A$ has global minimum at a single point $\boldsymbol{w}^* = 0$.

Let $\rho$ be a representation of $O(n)$ acting on $\mathbb{R}^n$. For $g \in O(n)$, we define the following group action:

$$g \cdot \boldsymbol{w} = A^{-\frac{1}{2}}\rho(g)A^{\frac{1}{2}}\boldsymbol{w}. \tag{15}$$

Then it can be shown that $\mathcal{L}_A(\boldsymbol{w})$ admits a $O(n)$ symmetry:

$$\mathcal{L}_A(g \cdot \boldsymbol{w}) = \boldsymbol{w}^T A^{\frac{1}{2}^T}\rho(g)^T A^{-\frac{1}{2}^T} A A^{-\frac{1}{2}}\rho(g)A^{\frac{1}{2}}\boldsymbol{w} = \boldsymbol{w}^T A \boldsymbol{w} = \mathcal{L}_A(\boldsymbol{w}). \tag{16}$$

Let $S_c = \{\boldsymbol{w} : \mathcal{L}_A(\boldsymbol{w}) = c\}$ be a level set of $\mathcal{L}_A$. We show that after a teleportation, every point on the gradient flow trajectory is optimal in its level set (with full proof in Appendix C.3).

**Proposition 5.5.** *If at point $\boldsymbol{w}$, $\|\nabla\mathcal{L}_A|_{\boldsymbol{w}}\|^2$ is at a maximum in $S_{\mathcal{L}_A(\boldsymbol{w})}$, then for any point $\boldsymbol{w}'$ on the gradient flow trajectory starting from $\boldsymbol{w}$, $\|\nabla\mathcal{L}_A|_{\boldsymbol{w}'}\|^2$ is at a maximum in $S_{\mathcal{L}_A(\boldsymbol{w}')}$.*

Finally, we observe that for $\mathcal{L}_A$, teleportation moves the parameters closer to the global minimum in Euclidean distance. In other words, maximizing the magnitude of gradient minimizes the distance to $\boldsymbol{w}^*$ in a loss level set.

**Proposition 5.6.** *Consider a point $\boldsymbol{w}$ in the parameter space. Let $g = \arg\max_{g \in G}\|\nabla\mathcal{L}_A|_{g \cdot \boldsymbol{w}}\|_2^2$. Then $g \cdot \boldsymbol{w} = \arg\min_{\boldsymbol{w}' \in S_{\mathcal{L}_A(\boldsymbol{w})}}\|\boldsymbol{w}' - \boldsymbol{w}^*\|_2^2$.*

## 5.4 Relation to second-order optimization methods

Since our algorithm involves optimizing the gradients, symmetry teleportation is closely related to second-order optimization methods. At the target point to teleport to, gradient descent becomes equivalent to Newton's method (proof in Appendix C.4).

**Proposition 5.7.** *Let $S_{\mathcal{L}_0} = \{\boldsymbol{w} : \mathcal{L}(\boldsymbol{w}) = \mathcal{L}_0\}$ be a level set of $\mathcal{L}$. If at a particular $\boldsymbol{w} \in S_{\mathcal{L}_0}$ we have $\|\nabla\mathcal{L}(\boldsymbol{w})\|_2 \geq \|\nabla\mathcal{L}(\boldsymbol{w}')\|_2$ for all $\boldsymbol{w}'$ in a small neighborhood of $\boldsymbol{w}$ in $S_{\mathcal{L}_0}$, then the gradient $\nabla\mathcal{L}(\boldsymbol{w})$ has the same direction as the direction from Newton's method, $H^{-1}\nabla\mathcal{L}(\boldsymbol{w})$.*

Proposition 5.7 provides an alternative way to interpret teleportation. Instead of computing the Newton's direction, we search within the loss level set for a point where the gradient has the same

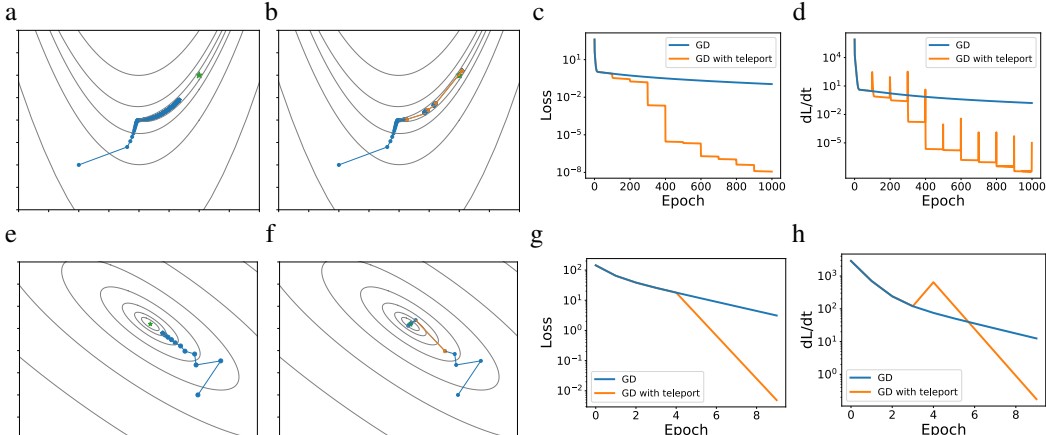

Figure 2: Optimization of the Rosenbrock function (top row) and Booth function (bottom row) using (a) gradient descent and (b) the proposed algorithm. Contours represent the level sets of the loss function. Loss $\mathcal{L}$ and convergence rate $d\mathcal{L}/dt$ are shown in (c) and (d). Teleportation helps move the parameters towards the target.

direction as Newton's direction. However, symmetry teleportation does not require computing the full Hessian matrix. The second derivative required for optimizing over continuous groups is obtained by taking derivatives with respect to parameters and then with respect to the group element, as opposed to taking the derivative with respect to parameters twice. This makes the computation significantly more feasible than Newton's method on neural networks with large number of parameters.

In practice, however, the group action used for teleportation is usually not transitive. Additionally, we do not teleport using the optimal group element since it can be unbounded. We also do not apply teleportation in every gradient descent step. Therefore, Proposition 5.7 serves as an intuition instead of an exact formulation of how teleportation works. We provide empirical evidence in Section 6 that these approximations do not erase the benefits of teleportation completely, and leave theoretical investigations of the connection to second-order methods under approximations as future work.

## 6   Experiments

### 6.1   Acceleration through symmetry teleportation

We examine the effect of symmetry teleportation on optimization. We illustrate teleportation in the parameter space on two test functions and show a speedup in regression and classification problems using multilayer neural networks. For test functions, we compared with GD for illustration purposes. For multi-layer neural networks, we also include AdaGrad as a more competitive baseline.

**Rosenbrock function.**   We apply symmetry teleportation to optimize the 2-variable Rosenbrock function (3). The parameters $x_1, x_2$ are initialized to $(-1, -1)$. Each algorithm is run 1000 steps with learning rate $10^{-3}$. We teleport the parameters every 100 steps. The group elements are found by gradient ascent on $\theta$, the parameter for the SO(2) group, for 10 steps with learning rate $10^{-1}$.

The trajectory of parameters and the loss level sets are plotted in Figure 2a,b. The blue star denotes the final position of parameters, the green star denotes the target, and orange dots are the positions from which symmetry transforms start. While gradient descent is not able to reach the target in 1000 steps, teleportation allows large steps and reaches the target much earlier. Figure 2d shows that teleportation improves the norm of gradients in the following step, and 2c shows its effect on the loss value. Teleportations clearly reduce the number of steps needed for convergence.

**Booth function.**   We also test symmetry teleportation on the Booth function defined in Eqn. (4). We initialize the parameters $x_1, x_2$ to $(5, -5)$. Each algorithm is run 10 steps with learning rate 0.08. We perform symmetry teleportation on the parameters once, before epoch 5. The group elements are found by gradient ascent on $\theta$ for 10 steps with learning rate 0.001. $\theta$ is initialized uniformly

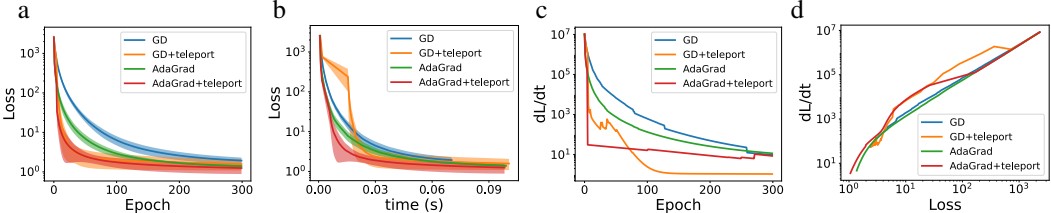

Figure 3: Multi-layer network optimization using gradient descent with and without teleportation. Teleportation reduces both the number of epochs and the total computational time required to reach convergence. At the same loss values, the teleported version has a larger gradient.

at random over $[0, \pi)$. Similar to the Rosenbrock function, teleportation moves the parameters to a trajectory with a larger convergence rate (Figure 2 bottom row).

**Multilayer neural network regression.** We further evaluated our method on a three-layer neural network with a regression loss $\min_{W_1,W_2,W_3} \|Y - W_3\sigma(W_2\sigma(W_1X))\|_2$. The dimension of weight matrices are $W_3 \in \mathbb{R}^{8\times7}$, $W_2 \in \mathbb{R}^{7\times6}$, and $W_1 \in \mathbb{R}^{6\times5}$. $X \in \mathbb{R}^{5\times4}$ is the data, $Y \in \mathbb{R}^{8\times4}$ is the target, and $\sigma$ is the LeakyReLU activation with slope coefficient 0.1. Additional hyperparameter details can be found in Appendix D.2.

Teleportation of the $\mathrm{GL}(\mathbb{R})$ group can be performed by finding an element $x$ in its Lie algebra, such that transforming $\boldsymbol{w}$ by the group element $g = \exp(x)$ improves the gradient $\frac{d\mathcal{L}}{dt}$. We use the first order approximation of the exponential map. Between each pair of weight matrices, we replace $g_m$ by $I + T$ and $g_m^{-1}$ by $I - T$ in (8), where $T \in \mathbb{R}^{d_m \times d_m}$ is initialized to 0. Then we perform gradient ascent steps on $T$ with objective defined in Line 3 of Algorithm 1 and update the pair of weights.

Figure 3a and 3b show the training curves plotted against epochs and time. Shaded area denotes one standard deviation from 5 runs. Since GD and AdaGrad use different learning rates, they are not directly comparable. However, the addition of teleportation clearly improves both algorithms. Figure 3c and 3d shows the squared norm of gradient from a single run. Teleportation increases the magnitude of gradient, and the trajectory with teleportation has a larger $d\mathcal{L}/dt$ value at the same loss values, which demonstrates that teleportation finds a better trajectory.

**MNIST classification.** We apply symmetry teleportation on the MNIST classification task (Deng, 2012). We split the training set into 48,000 for training and 12,000 for validation. The input data has dimension $28 \times 28$ and is flattened into a vector. The output of the neural network has dimension 10 corresponding to the 10 digit classes. We used a three-layer neural network with hidden dimension $[512, 512]$, LeakyReLU activations, and cross-entropy loss. Learning rate is $2 \times 10^{-3}$, and learning rate for teleportation is $10^{-3}$. Each optimization algorithm is run 80 epochs with batch size of 20. Immediately after the first epoch, we apply teleportation using data from one mini-batch, and repeat for 4 different mini-batches. For each mini-batch, 10 gradient ascent steps are used to optimize $g_m$.

Figure 4a,b shows the effect of teleportation on training and validation loss, and Figure 4c,d shows the norm of the gradient in training. While teleportation significantly accelerates the decrease of the training loss in SGD, its effect on the validation loss is limited and detrimental for AdaGrad. Therefore, teleportation on MNIST makes training faster at the beginning but leads to earlier overfit and slightly worse validation accuracy. A possible reason is that regions with large gradients have sharp minima that do not generalize well.

## 6.2 Teleportation schedule

The effect of teleportation varies depending on its time and frequency. Figure 5 shows the result of teleportation on MNIST with different hyperparameters. In all experiments, we use SGD with batch size 20, learning rate $2 \times 10^{-3}$, and 10 gradient ascent steps for each teleportation.

In Figure 5a, we randomly select 4 different mini-batches and apply teleportation on each of them individually, but at different epochs. Teleportation before training has the worst performance. After epoch 0, the effect of teleportation is stronger when it is applied earlier. In Figure 5b, we again apply teleportations on 4 randomly selected mini-batches, but repeat this with 5 different teleportation

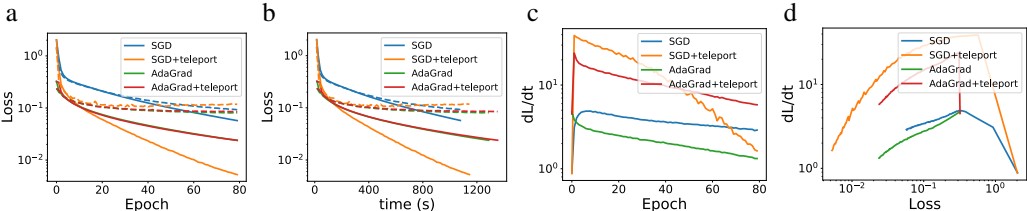

Figure 4: MNIST classification using gradient descent with and without teleportation. Solid lines are training loss and dashed lines are validation loss.

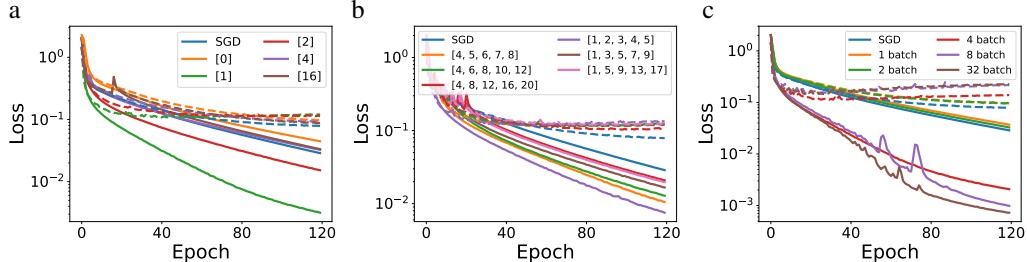

Figure 5: Teleportation (a) once at different epoch, (b) 5 times with different teleportation schedules, and (c) using different number of mini-batches. The lists in the legend of (a) and (c) denote the epoch numbers in teleportation schedule where teleportation happens.

schedules (hyperparameter $K$ in Algorithm 2) as shown in the legend. All schedules have the same number of teleportations. Smaller intervals between teleportations accelerate convergence more significantly. In Figure 5c, we apply teleportation immediately after the first epoch, but use different numbers of mini-batches (hyperparameter $B$ in Algorithm 2) and teleport using each of them individually. Using more mini-batches to teleport leads to faster decrease in training loss but is also more prone to overfitting.

## 6.3 Runtime analysis

The additional amount of computation introduced by teleportation depends on the implementation of Line 2-5 of Algorithm 1. We discuss our implementation for teleporting multi-layer neural networks as an example. Teleporting each pair of adjacent weight matrices requires computing the inverse of the output from the previous layer. Denote the batch size as $n$, the largest dimension of all layers as $d_{max}$, and the number of layers as $l$. Assume that $d_{max} > n$. Computing the inverse of the output of each layer has complexity $O(d_{max}^2 n)$, and computing the pseudoinverse for all layers has complexity $O(d_{max}^2 nl)$. Note that all matrices we invert have dimensions at most $d_{max} \times n$.

For one gradient ascent step on $g$, the forward and backward pass both have complexity $O(d_{max}^2 nl)$. This is the same as the forward and backward pass of gradient descent on $\boldsymbol{w}$ because the architecture is the same except with approximately twice as many layers. We perform $t$ gradient ascent steps on $g$. Therefore, the computation cost for one teleportation is $O(d_{max}^2 nlt)$.

We show empirically that the runtime for teleportation scales polynomially with matrix dimensions and linearly with the number of layers. We record the runtime of gradient descent on a Leaky-ReLU neural network for 300 epochs, with a 10-step teleportation every 10 epochs. Each pair of adjacent weight matrices is transformed by a group action during teleportation. Figure 6(a) shows the runtime for a 3-layer network using square weights and data matrices with different dimensions. Figure 6(b) shows the runtime for 128-by-128 weight and data matrices with different number of layers.

Although the teleportation step has the same complexity as a gradient descent step, the runtime is dominated by teleportation due to larger constants in the complexity analysis. The trade-off between teleportation time and convergence rate depends on specific problems. In our experiments, the convergence rate is improved by a small number of teleportation steps which does not add significant computational overhead.

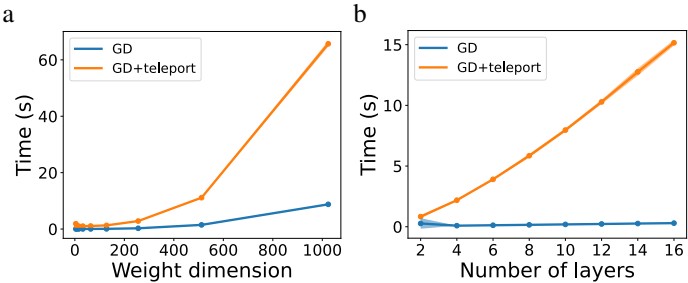

Figure 6: Gradient descent training time on a Leaky-ReLU neural network for 300 epochs, with a 10-step teleportation every 10 epochs. (a) 3-layer network using square weights and data matrices, with different dimensions. (b) Weights and data are all 128 by 128 matrices, but the network has different number of layers.

# 7 Discussion and Conclusions

We proposed a new optimization algorithm that exploits the symmetry in the loss landscape to improve convergence. It is interesting to note that optimizing $d\mathcal{L}/dt$ locally sometimes leads to an improved global path. One example is the ellipse function (Figure 1), where teleporting once ensures that $d\mathcal{L}/dt$ is optimal at every $\mathcal{L}$ value along the trajectory. Another example is the matrix factorization problem $\mathcal{L}(U, V) = \|Y - UV\|_2^2$. Tarmoun et al. (2021) shows that the convergence rate increases with the imbalance $U^T U - V^T V$. Consider the transformation $U, V \to Ug, g^{-1}V$. To optimize $d\mathcal{L}(U, V)/dt$ locally, we would need a large $g \in \mathrm{GL}_n$, but a large $g$ also leads to a large $U^T U - V^T V$ which is positively correlated with the overall convergence rate of the entire trajectory. Hence teleportation is guaranteed to produce a better trajectory.

A potential future direction is to derive the exact expression for how teleportation affects the loss value at a later time in gradient flow, which may lead to a closed-form solution of the optimal teleportation destination. Additionally, inspired by the landscape view of parameter space symmetry Şimşek et al. (2021), teleportation using discrete (permutation) symmetries may allow us to reach a better minimum. Finally, additional theory can be developed to explain the relationship between teleportation and second-order methods under the approximations we introduced, especially to quantify the improvement on the overall convergence rate, and to derive its effect on generalization bounds. Integrating teleportation with other advanced optimizers such as Adam and RMSprop would be another interesting future step.

## Acknowledgments and Disclosure of Funding

This work was supported in part by U.S. Department Of Energy, Office of Science, U. S. Army Research Office under Grant W911NF-20-1-0334, Google Faculty Award, Amazon Research Award, and NSF Grants #2134274, #2107256 and #2134178. R. Walters is supported by the Roux Institute and the Harold Alfond Foundation. We are grateful to Iordan Ganev for many helpful discussions.

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
