# A Adaptations of Algorithm 1 for different problems

## A.1 Stochastic gradient descent

We extend Algorithm 1 to stochastic gradient descent (SGD). We apply the group actions using data from a mini-batch $X_i$, and repeat for $B$ mini-batches each time. The gradient we optimize, $\tilde{\nabla}\mathcal{L}(X_i, g \cdot \boldsymbol{w}_t)$, also uses single mini-batches. Algorithm 2 provides the framework for teleportation in SGD.

---

**Algorithm 2:** Symmetry Teleportation (SGD)

---

**Input:** Loss function $\mathcal{L}(\boldsymbol{w})$, learning rate $\eta$, number of epochs $t_{max}$, initialized parameters $\boldsymbol{w}_0$, symmetry group $G$, teleportation schedule $K$, number of mini-batches used to teleport $B$.

**Output:** $\boldsymbol{w}_{t_{max}}$.

1 **for** $t \leftarrow 0$ **to** $t_{max} - 1$ **do**
2    **if** $t \in K$ **then**
3      **for** $X_i$ *in the first $B$ mini-batches* **do**
4        $g \leftarrow \operatorname{argmax}_{g \in G} \|\tilde{\nabla}\mathcal{L}(X_i, g \cdot \boldsymbol{w}_t)\|^2$
5        $\boldsymbol{w}_t \leftarrow g \cdot \boldsymbol{w}_t$
6        $\boldsymbol{w}_t \leftarrow \boldsymbol{w}_t - \eta\tilde{\nabla}\mathcal{L}(X_i, \boldsymbol{w}_t)$
7      **end for**
8      **for** $X_i$ *in the rest mini-batches* **do**
9        $\boldsymbol{w}_t \leftarrow \boldsymbol{w}_t - \eta\tilde{\nabla}\mathcal{L}(X_i, \boldsymbol{w}_t)$
10      **end for**
11    **else**
12      **for** *all mini-batches* $X_i$ **do**
13        $\boldsymbol{w}_t \leftarrow \boldsymbol{w}_t - \eta\tilde{\nabla}\mathcal{L}(X_i, \boldsymbol{w}_t)$
14      **end for**
15    **end if**
16    $\boldsymbol{w}_{t+1} \leftarrow \boldsymbol{w}_t$
17 **end for**
18 **return** $\boldsymbol{w}_{t_{max}}$

---

## A.2 Data transformation

Algorithm 3 here modifies Algorithm 1 to allow transformations on both parameters and data. Denote $g_X$ as the group action on data only. The group actions on data at all teleportation steps can be precomposed as a function $f$ and applied to the input data at inference time.

---

**Algorithm 3:** Symmetry Teleportation (with data transformation)

---

**Input:** Loss function $\mathcal{L}(\boldsymbol{w}, X)$, learning rate $\eta$, number of epochs $t_{max}$, initialized parameters $\boldsymbol{w}_0$, symmetry $G$, teleportation schedule $K$, data $X$.

**Output:** $\boldsymbol{w}_{t_{max}}$, data transformation $f$.

**Initialize:** $f$ = the identity function.

1 **for** $t \leftarrow 0$ **to** $t_{max} - 1$ **do**
2    **if** $t \in K$ **then**
3      $g \leftarrow \operatorname{argmax}_{g \in G} \|\nabla_{g \cdot \boldsymbol{w}_t}\mathcal{L}(g \cdot (\boldsymbol{w}_t, X))\|^2$
4      $\boldsymbol{w}_t, X \leftarrow g \cdot (\boldsymbol{w}_t, X)$
5      $f \leftarrow g_X \circ f$
6    **end if**
7    $\boldsymbol{w}_{t+1} \leftarrow \boldsymbol{w}_t - \eta\nabla_{\boldsymbol{w}_t}\mathcal{L}$
8 **end for**
9 **return** $\boldsymbol{w}_{t_{max}}, f$

---

# B  Group actions

In this section, we derive the group actions for the test functions and multi-layer neural networks. More details about group theory can be found in textbooks such as Lang (2002).

## B.1  Continuous symmetry in test functions

### B.1.1  Ellipse

Consider the following loss function with $a \in \mathbb{R}^{\geq 0}$:

$$\mathcal{L}(x_1, x_2) = x_1^2 + ax_2^2 \tag{17}$$

If we change the variables to $\mathcal{L}(u(x_1, x_2), v(x_1, x_2)) = u^2 + v^2$, 2D rotations leave $\mathcal{L}$ unchanged. Therefore $SO(2)$ is a symmetry of $\mathcal{L}(x_1, x_2)$. Let $g_\theta \in SO(2)$, and define the group action as

$$g_\theta \cdot \begin{bmatrix} x_1 \\ x_2 \end{bmatrix} = \begin{bmatrix} 1 & 0 \\ 0 & 1/\sqrt{a} \end{bmatrix} \begin{bmatrix} \cos\theta & -\sin\theta \\ \sin\theta & \cos\theta \end{bmatrix} \begin{bmatrix} 1 & 0 \\ 0 & \sqrt{a} \end{bmatrix} \begin{bmatrix} x_1 \\ x_2 \end{bmatrix} \tag{18}$$

Then

$$\mathcal{L}(x_1, x_2) = \mathcal{L}(g \cdot (x_1, x_2)) \tag{19}$$

### B.1.2  Rosenbrock function

Consider the Rosenbrock function with 2 variables Rosenbrock (1960):

$$\mathcal{L}(x_1, x_2) = 100(x_1^2 - x_2)^2 + (x_1 - 1)^2 \tag{20}$$

Let $u = 10(x_1^2 - x_2)$ and $v = x_1 - 1$. After changing the variables from $x$ and $y$ to $u$ and $v$, $\mathcal{L}$ has a rotational symmetry. Note that the function, $h : \mathbb{R}^2 \to \mathbb{R}^2$, that maps $x_1, x_2$ to $u, v$ is bijective:

$$(u, v) = h(x_1, x_2) = (10(x_1^2 - x_2), x_1 - 1)$$
$$(x_1, x_2) = h^{-1}(u, v) = (v + 1, (v + 1)^2 - 0.1u)$$
$$h(x_1, x_2) = h(y_1, y_2) \Rightarrow (x_1, x_2) = (y_1, y_2) \tag{21}$$

Next, we show that $SO(2)$ is a symmetry of $\mathcal{L}(x_1, x_2)$. Let $\rho$ be a representation of $SO(2)$ acting on $\mathbb{R}^2$. For $g \in SO(2)$, define the following group action:

$$g \cdot (x_1, x_2) = h^{-1}\left(\rho(g)h(x_1, x_2)\right) \tag{22}$$

Then

$$\mathcal{L}(x_1, x_2) = \mathcal{L}(g \cdot (x_1, x_2)) \tag{23}$$

For the Rosenbrock function with 2N parameters, we can construct a bijective function $h : \mathbb{R}^{2N} \to \mathbb{R}^{2N}$ by transforming each of the $N$ pairs of variables as before, and $SO(2N)$ is a symmetry of $\mathcal{L}(x_1, ..., x_{2N})$. However, we will only use the 2 variable version in the experiments.

### B.1.3  Booth function

Consider the Booth function Jamil and Yang (2013):

$$\mathcal{L}(x_1, x_2) = (x_1 + 2x_2 - 7)^2 + (2x_1 + x_2 - 5)^2$$

Similar to the Rosebrock function, a change of variables reveals a rotational symmetry of $\mathcal{L}$:

$$(u, v) = h(x_1, x_2) = (x_1 + 2x_2 - 7, 2x_1 + x_2 - 5)$$
$$(x_1, x_2) = h^{-1}(u, v) = (-\frac{1}{3}u + \frac{2}{3}v + 1, \frac{2}{3}u - \frac{1}{3}v + 3)$$
$$\tag{24}$$

The function $h : \mathbb{R}^2 \to \mathbb{R}^2$ that maps $x_1, x_2$ to $u, v$ is bijective. Let $\rho$ be a representation of $SO(2)$ acting on $\mathbb{R}^2$. For $g \in SO(2)$, define the following group action:

$$g \cdot (x_1, x_2) = h^{-1}\left(\rho(g)h(x_1, x_2)\right) \tag{25}$$

Then $\mathcal{L}(x_1, x_2)$ admits an $SO(2)$ symmetry:

$$\mathcal{L}(x_1, x_2) = \mathcal{L}(g \cdot (x_1, x_2)) \tag{26}$$

## B.2 Continuous symmetry in multi-layer neural Networks

In this and the following sections, we provide proofs for the theoretical results. We restate the propositions from the main text for readability.

**Proposition 4.2.** *A linear network is invariant under all groups $G_m \equiv \mathrm{GL}_{d_m}$ acting as*

$$g \cdot (W_m, W_{m-1}) = (W_m g^{-1}, g W_{m-1}), \qquad g \cdot W_k = W_k, \quad \forall k \notin \{m, m-1\}.$$

*Proof.* In the linear network $h_m = W_m h_{m-1}$, hence

$$g \cdot (W_m, h_{m-1}) = (W_m g^{-1}, g h_{m-1}), \quad g \cdot h_m = W_m g^{-1} g h_{m-1} = h_m \tag{27}$$

which means a $p$-layer linear network is invariant under all $G_m$ with $m \leq p$ as they keep the output $h_p$ invariant ($\forall g \in G_m, g \cdot h_p = h_p$). $\qquad\square$

**Proposition 4.3.** *Assume that $h_{m-2}$ is invertible. A multi-layer network with bijective activation $\sigma$ has a $\mathrm{GL}_{d_{m-1}}$ symmetry. For $g_m \in G_m = \mathrm{GL}_{d_{m-1}}(\mathbb{R})$ the following group action keeps $h_p$ with $p \geq m$ invariant*

$$g_m \cdot W_k = \begin{cases} W_m g_m^{-1} & k = m \\ \sigma^{-1}\left(g_m \sigma\left(W_{m-1} h_{m-2}\right)\right) h_{m-2}^{-1} & k = m-1 \\ W_k & k \notin \{m, m-1\} \end{cases} \tag{28}$$

*Proof.* From (7), we want to convert $g \cdot h_{m-1}$ into a transformation on $W_{m-1}$ instead of $h_{m-1}$. In other words, we want to find a set of transformed weights $W'_m, W'_{m-1}$ which yields the same network output $\tilde{h}_m$:

$$\tilde{h}_m = W'_m \sigma\left(W'_{m-1} h_{m-2}\right) = W_m g^{-1} g \sigma\left(W_{m-1} h_{m-2}\right)$$
$$\Rightarrow W'_m = W_m g^{-1}, \qquad \sigma\left(W'_{m-1} h_{m-2}\right) = g \sigma\left(W_{m-1} h_{m-2}\right) \tag{29}$$

Solving (29) we get

$$W'_{m-1} = \sigma^{-1}\left(g \sigma\left(W_{m-1} h_{m-2}\right)\right) h_{m-2}^{-1}. \tag{30}$$

(28) follows from (29) and (30).

To verify that (28) is a valid group action,

$$I \cdot W_k = \begin{cases} W_m I & k = m \\ \sigma^{-1}\left(I \sigma\left(W_{m-1} h_{m-2}\right)\right) h_{m-2}^{-1} & k = m-1 \\ W_k & k \notin \{m, m-1\} \end{cases}$$
$$= W_k \tag{31}$$

and

$$g_1 \cdot (g_2 \cdot W_k) = \begin{cases} W_m g_2^{-1} g_1^{-1} & k = m \\ \sigma^{-1}\left(g_1 \sigma\left(\left[\sigma^{-1}\left(g_2 \sigma\left(W_{m-1} h_{m-2}\right)\right) h_{m-2}^{-1}\right] h_{m-2}\right)\right) h_{m-2}^{-1} & k = m-1 \\ W_k & k \notin \{m, m-1\} \end{cases}$$
$$= \begin{cases} W_m (g_1 g_2)^{-1} & k = m \\ \sigma^{-1}\left((g_1 g_2) \sigma\left(W_{m-1} h_{m-2}\right)\right) h_{m-2}^{-1} & k = m-1 \\ W_k & k \notin \{m, m-1\} \end{cases}$$
$$= (g_1 g_2) \cdot W_k$$

$$\tag{32}$$

$\qquad\square$

# C  Theoretical analysis of teleportation

## C.1  What symmetries help accelerate optimization

**Proposition 5.1.** *Let $\boldsymbol{w}' = g \cdot \boldsymbol{w}$ be a point we teleport to. Let $J = \partial \boldsymbol{w}'/\partial \boldsymbol{w}$ be the Jacobian. Symmetry teleportation using $g$ accelerates the rate of decay in $\mathcal{L}$ if it satisfies*

$$\left\| \left[J^{-1}\right]^T \nabla \mathcal{L}(\boldsymbol{w}) \right\|_\eta^2 > \|\nabla \mathcal{L}(\boldsymbol{w})\|_\eta^2.$$

*Proof.* Let $\boldsymbol{w}' = g \cdot \boldsymbol{w}$. Denote the Jacobian as $J$, where $J_{ij} = \partial w_i'/\partial w_j$. Then the inverse of $J$ has entries $J_{ij}^{-1} = \partial w_i/\partial w_j'$.

The gradient at $\boldsymbol{w}'$ is

$$\frac{\partial \mathcal{L}(\boldsymbol{w}')}{\partial \boldsymbol{w}'} = \frac{\partial \mathcal{L}(\boldsymbol{w})}{\partial \boldsymbol{w}'} = \sum_j \frac{\partial \mathcal{L}(\boldsymbol{w})}{\partial \boldsymbol{w}_j} \frac{\partial \boldsymbol{w}_j}{\partial \boldsymbol{w}_i'} = \sum_j \frac{\partial \mathcal{L}(\boldsymbol{w})}{\partial \boldsymbol{w}_j} J_{ji}^{-1} = \left( \left( \frac{\partial \mathcal{L}(\boldsymbol{w})}{\partial \boldsymbol{w}} \right)^T J^{-1} \right)^T = (J^{-1})^T \frac{\partial \mathcal{L}(\boldsymbol{w})}{\partial \boldsymbol{w}} \tag{33}$$

The rate of change of $\mathcal{L}$ in gradient flow is

$$\frac{d\mathcal{L}(\boldsymbol{w}')}{dt} = \left\langle \frac{\partial \mathcal{L}}{\partial \boldsymbol{w}'}, \frac{d\boldsymbol{w}'}{dt} \right\rangle = -\left\| (J^{-1})^T \frac{\partial \mathcal{L}(\boldsymbol{w})}{\partial \boldsymbol{w}} \right\|_\eta^2 \tag{34}$$

Thus we will have a speedup if

$$\left\| (J^{-1})^T \frac{\partial \mathcal{L}(\boldsymbol{w})}{\partial \boldsymbol{w}} \right\|_\eta^2 > \left\| \frac{\partial \mathcal{L}(\boldsymbol{w})}{\partial \boldsymbol{w}} \right\|_\eta^2 \tag{35}$$

$\square$

**Proof of Corollary 5.2**

*Proof.* Since $J = g$, using (9) and the l.h.s. of (10) we have

$$\frac{d\mathcal{L}(g \cdot \boldsymbol{w})}{dt} = -\nabla \mathcal{L}^T g^{-1} \eta \left[g^{-1}\right]^T \nabla \mathcal{L} = \nabla \mathcal{L}^T \left[g^T \eta^{-1} g\right]^{-1} \nabla \mathcal{L} = \|\nabla \mathcal{L}\|_\eta^2 \tag{36}$$

$\square$

## C.2  Improvement of subsequent steps

**Proposition 5.4.** *Consider the gradient descent with a $G$-invariant loss $\mathcal{L}(\boldsymbol{w})$ and learning rate $\eta \in \mathbb{R}^+$. Let $\boldsymbol{w}_t$ be the parameter at time $t$ and $\boldsymbol{w}_t' = g \cdot \boldsymbol{w}_t$ the parameter teleported by $g \in G$. Let $\boldsymbol{w}_{t+T}$ and $\boldsymbol{w}_{t+T}'$ be the parameters after $T$ more steps of gradient descent from $\boldsymbol{w}_t$ and $\boldsymbol{w}_t'$ respectively. Under Assumption 5.3, if $\eta L < 1$, and*

$$\frac{\|\partial \mathcal{L}/\partial \boldsymbol{w}_t'\|_2}{\|\partial \mathcal{L}/\partial \boldsymbol{w}_t\|_2} \geq \frac{(1+\eta L)^T}{(1-\eta L)^T},$$

*then*

$$\left\| \frac{\partial \mathcal{L}}{\partial \boldsymbol{w}_{t+T}'} \right\|_2 \geq \left\| \frac{\partial \mathcal{L}}{\partial \boldsymbol{w}_{t+T}} \right\|_2.$$

*Proof.* From the definition of Lipschitz continuity and the update rule of gradient descent,

$$\left| \left\| \frac{\partial \mathcal{L}}{\partial \boldsymbol{w}_{t+1}} \right\|_2 - \left\| \frac{\partial \mathcal{L}}{\partial \boldsymbol{w}_t} \right\|_2 \right| \leq L \|\boldsymbol{w}_{t+1} - \boldsymbol{w}_t\|_2 = L \left\| \eta \frac{\partial \mathcal{L}}{\partial \boldsymbol{w}_t} \right\|_2 \tag{37}$$

Equivalently,

$$(1 - \eta L) \left\| \frac{\partial \mathcal{L}}{\partial \boldsymbol{w}_t} \right\|_2 \leq \left\| \frac{\partial \mathcal{L}}{\partial \boldsymbol{w}_{t+1}} \right\|_2 \leq (1 + \eta L) \left\| \frac{\partial \mathcal{L}}{\partial \boldsymbol{w}_t} \right\|_2 \tag{38}$$

By unrolling $T$ steps, we have

$$(1 - \eta L)^T \left\| \frac{\partial \mathcal{L}}{\partial \boldsymbol{w}_t} \right\|_2 \leq \left\| \frac{\partial \mathcal{L}}{\partial \boldsymbol{w}_{t+T}} \right\|_2 \leq (1 + \eta L)^T \left\| \frac{\partial \mathcal{L}}{\partial \boldsymbol{w}_t} \right\|_2 \tag{39}$$

Similarly, for a teleported point $\boldsymbol{w}'_t = g \cdot \boldsymbol{w}_t$,

$$(1 - \eta L)^T \left\| \frac{\partial \mathcal{L}}{\partial \boldsymbol{w}'_t} \right\|_2 \leq \left\| \frac{\partial \mathcal{L}}{\partial \boldsymbol{w}'_{t+T}} \right\|_2 \leq (1 + \eta L)^T \left\| \frac{\partial \mathcal{L}}{\partial \boldsymbol{w}'_t} \right\|_2 \tag{40}$$

Therefore, if

$$(1 - \eta L)^T \left\| \frac{\partial \mathcal{L}}{\partial \boldsymbol{w}'_t} \right\|_2 \geq (1 + \eta L)^T \left\| \frac{\partial \mathcal{L}}{\partial \boldsymbol{w}_t} \right\|_2 \tag{41}$$

then it is guaranteed that

$$\left\| \frac{\partial \mathcal{L}}{\partial \boldsymbol{w}'_{t+T}} \right\|_2 \geq \left\| \frac{\partial \mathcal{L}}{\partial \boldsymbol{w}_{t+T}} \right\|_2 \tag{42}$$

$\square$

### C.3 Convergence analysis for convex quadratic functions

We first show that starting from a point in $S_c$, all other points in $S_c$ can be reached with one teleportation.

**Proposition C.1.** $S_c$ *contains a single orbit. That is,* $G \cdot \boldsymbol{w} \equiv \{g \cdot \boldsymbol{w} : g \in G\} = S_c$ *for all* $\boldsymbol{w} \in S_c$.

*Proof.* Consider two points $\boldsymbol{w}_1, \boldsymbol{w}_2 \in S_c$. Then $\boldsymbol{w}_1^T A \boldsymbol{w}_1 = (A^{\frac{1}{2}} \boldsymbol{w}_1)^T (A^{\frac{1}{2}} \boldsymbol{w}_1) = c$ and $\boldsymbol{w}_2^T A \boldsymbol{w}_2 = (A^{\frac{1}{2}} \boldsymbol{w}_2)^T (A^{\frac{1}{2}} \boldsymbol{w}_2) = c$. Let $\boldsymbol{v}_1 = \frac{A^{\frac{1}{2}} \boldsymbol{w}_1}{\sqrt{c}}$, $\boldsymbol{v}_2 = \frac{A^{\frac{1}{2}} \boldsymbol{w}_2}{\sqrt{c}}$ and $\mathbf{e}_1 = [1, 0, ..., 0]^T$. Since $\|v_1\| = \|v_2\| = 1$, there exists $g_1, g_2 \in O(n)$, such that $g_1 \mathbf{e}_1 = \boldsymbol{v}_1$ and $g_2 \mathbf{e}_1 = \boldsymbol{v}_2$. One way to construct such $g_1$ is let the first column be equal to $\boldsymbol{v}_1$ and other columns be the rest of the orthonormal basis. Let $g = g_2 g_1^{-1}$. Then $\boldsymbol{v}_2 = g \boldsymbol{v}_1$, $A^{\frac{1}{2}} \boldsymbol{w}_2 = g A^{\frac{1}{2}} \boldsymbol{w}_1$, and $\boldsymbol{w}_2 = A^{-\frac{1}{2}} g A^{\frac{1}{2}} \boldsymbol{w}_1 = g \cdot \boldsymbol{w}_1$.

We have shown that for any $\boldsymbol{w}_1, \boldsymbol{w}_2 \in S_c$, there exists a $g \in G$ such that $\boldsymbol{w}_2 = g \cdot \boldsymbol{w}_1$. Therefore, the group action of $G$ on $S_c$ is transitive. Equivalently, $S_c$ contains a single orbit. $\square$

The objective of teleportation is transforming parameters using a group element to maximize the norm of gradient:

$$\max_{g \in G} \|\nabla \mathcal{L}_A|_{g \cdot \boldsymbol{w}}\|_2^2. \tag{43}$$

Since all points on the level set are reachable, the target teleportation destination is the point with the largest gradient norm on the same level set. In other words, (43) is equivalent to the following optimization problem:

$$\max_{\boldsymbol{w}'} \|\nabla \mathcal{L}_A|_{\boldsymbol{w}'}\|_2^2$$
$$\text{s.t. } \mathcal{L}_A(\boldsymbol{w}') = \mathcal{L}_A(\boldsymbol{w}). \tag{44}$$

Let $c = \mathcal{L}_A(\boldsymbol{w})$. Substitute in $\mathcal{L}_A$ and $\nabla \mathcal{L}_A$, we have the following equivalent formulation:

$$\max_{\boldsymbol{w}'} \|A \boldsymbol{w}'\|_2^2$$
$$\text{s.t. } \boldsymbol{w}'^T A \boldsymbol{w}' = c. \tag{45}$$

Next, we solve this optimization problem and show that the gradient norm is maximized on the gradient flow trajectory starting from its solution.

**Proposition C.2.** *The solution to (45) is an eigenvector of $A$ corresponding to its largest eigenvalue.*

*Proof.* We solve (45) using the method of Lagrangian multipliers. The Lagrangian of (45) is

$$\mathcal{L} = \boldsymbol{w}'^T A^T A \boldsymbol{w}' - \lambda(\boldsymbol{w}'^T A \boldsymbol{w}' - c). \tag{46}$$

Setting the derivative with respect of $\boldsymbol{w}'$ to 0, we have

$$\frac{\partial \mathcal{L}}{\partial \boldsymbol{w}} = 2A^T A \boldsymbol{w}' - 2\lambda A \boldsymbol{w}' = 0, \tag{47}$$

which gives

$$A^T A \boldsymbol{w}' = \lambda A \boldsymbol{w}'. \tag{48}$$

Since $A$ is positive definite, $A = A^T$ and $A$ is invertible. Therefore,

$$A \boldsymbol{w}' = \lambda \boldsymbol{w}', \tag{49}$$

so the solution to (45) is an eigenvector of $A$. Then, the constraint is $\boldsymbol{w}'^T A \boldsymbol{w}' = \lambda \|\boldsymbol{w}'\|^2 = c$, and the objective becomes $\max_{\boldsymbol{w}'} \lambda^2 \|\boldsymbol{w}'\|^2 = \max_{\boldsymbol{w}'} c\lambda$. Therefore, we want $\lambda$ to be the largest eigenvalue of $A$. Hence the optimal $\boldsymbol{w}'$ is an eigenvector of $A$ corresponding to its largest eigenvalue. $\square$

**Proposition 5.5.** *If at point $\boldsymbol{w}$, $\|\nabla \mathcal{L}_A|_{\boldsymbol{w}}\|^2$ is at a maximum in $S_{\mathcal{L}_A(\boldsymbol{w})}$, then for any point $\boldsymbol{w}'$ on the gradient flow trajectory starting from $\boldsymbol{w}$, $\|\nabla \mathcal{L}_A|_{\boldsymbol{w}'}\|^2$ is at a maximum in $S_{\mathcal{L}_A(\boldsymbol{w}')}$.*

*Proof.* From Proposition C.2, $\boldsymbol{w}$ is an eigenvector of $A$ corresponding to its largest eigenvalue. Then the gradient of $\mathcal{L}_A$ is $A\boldsymbol{w} = \lambda\boldsymbol{w}$. Therefore, on the gradient flow trajectory starting from $\boldsymbol{w}$, every point has the same direction as $\boldsymbol{w}$, meaning that the points are all eigenvectors of $A$ corresponding to its largest eigenvalue. Therefore, $\|\nabla \mathcal{L}_A\|^2$ is always at a maximum on the loss level sets. $\square$

Finally, we show that maximizing the magnitude of gradient is equivalent to minimizing the distance to $\boldsymbol{w}^*$ in a loss level set (Proposition 5.6).

**Proposition C.3.** *The solution to the following optimization problem is the same as the solution to (45):*

$$\min_{\boldsymbol{w}'} \|\boldsymbol{w}' - \boldsymbol{w}^*\|_2^2$$
$$s.t. \ \boldsymbol{w}'^T A \boldsymbol{w}' = c. \tag{50}$$

*Proof.* Similar to Proposition C.2, we solve this optimization problem using the method of Lagrangian multipliers. Note that $\boldsymbol{w}^* = 0$. The Lagrangian is

$$\mathcal{L} = \boldsymbol{w}'^T \boldsymbol{w}' - \lambda(\boldsymbol{w}'^T A \boldsymbol{w}' - c). \tag{51}$$

Setting the derivative with respect to $\boldsymbol{w}'$ to 0, we have

$$\frac{\partial \mathcal{L}}{\partial \boldsymbol{w}} = 2\boldsymbol{w}' - 2\lambda A \boldsymbol{w}' = 0, \tag{52}$$

which gives

$$A \boldsymbol{w}' = \lambda \boldsymbol{w}', \tag{53}$$

so the solution to (50) is an eigenvector of $A$. Then, the constraint is $\boldsymbol{w}'^T A \boldsymbol{w}' = \lambda \|\boldsymbol{w}'\|^2 = c$, and the objective becomes $\min_{\boldsymbol{w}'} \|\boldsymbol{w}'\|^2 = \min_{\boldsymbol{w}'} \frac{c}{\lambda}$. Therefore, we want $\lambda$ to be the largest eigenvalue of $A$. Hence the optimal $\boldsymbol{w}'$ is an eigenvector of $A$ corresponding to its largest eigenvalue, which is the same as the solution to (45). $\square$

For a more concrete example, consider a diagonal matrix $A$ with positive diagonal elements. Then the level sets of $\mathcal{L}_A$ are $n$-dimensional ellipsoids centered at the origin 0, with axes in the same directions as the standard basis. The point with largest $\|\nabla \mathcal{L}_A\|^2$ on a level set is in the eigendirection of $A$ corresponding to its largest eigenvalue, or equivalently, a point on the smallest semi-axes of the ellipsoid. Note that this point has the smallest distance to the global minimum $\boldsymbol{w}^* = 0$ among all points in the same level set. In addition, the gradient flow trajectory from this point always points to $\boldsymbol{w}^*$. Therefore, like the 2D ellipse function, one teleportation on the $n$-dimensional ellipsoid also guarantees optimal gradient norm at all points on the trajectory.

## C.4 Relation to second-order optimization methods

To prove Proposition 5.7, we first note that when the norm of the gradient is at a critical point on the level set of the loss function, the gradient is an eigenvector of the Hession.

**Lemma C.4.** *If* $\partial_{\boldsymbol{v}} \left\| \frac{\partial \mathcal{L}}{\partial \boldsymbol{w}} \right\|_2^2 = 0$ *for all unit vector* $\boldsymbol{v}$ *that is orthogonal to* $\frac{\partial \mathcal{L}}{\partial \boldsymbol{w}}$*, then* $\frac{\partial \mathcal{L}}{\partial \boldsymbol{w}}$ *is an eigenvector of the Hessian of* $\mathcal{L}$*.*

*Proof.* From the definition of the directional derivative,

$$\partial_{\boldsymbol{v}} \left\| \frac{\partial \mathcal{L}}{\partial \boldsymbol{w}} \right\|_2^2 = \boldsymbol{v} \cdot \frac{\partial}{\partial \boldsymbol{w}} \left\| \frac{\partial \mathcal{L}}{\partial \boldsymbol{w}} \right\|_2^2 \tag{54}$$

Writing the last term in indices,

$$\frac{\partial}{\partial \boldsymbol{w}_i} \left\| \frac{\partial \mathcal{L}}{\partial \boldsymbol{w}} \right\|_2^2 = \frac{\partial}{\partial \boldsymbol{w}_i} \sum_j \left( \frac{\partial \mathcal{L}}{\partial \boldsymbol{w}_j} \right)^2$$

$$= \sum_j \frac{\partial}{\partial \boldsymbol{w}_i} \left( \frac{\partial \mathcal{L}}{\partial \boldsymbol{w}_j} \right)^2$$

$$= \sum_j 2 \frac{\partial \mathcal{L}}{\partial \boldsymbol{w}_j} \frac{\partial^2 \mathcal{L}}{\partial \boldsymbol{w}_i \partial \boldsymbol{w}_j}$$

$$= 2 \left( H \frac{\partial \mathcal{L}}{\partial \boldsymbol{w}} \right)_i \tag{55}$$

Removing the indices,

$$\frac{\partial}{\partial \boldsymbol{w}} \left\| \frac{\partial \mathcal{L}}{\partial \boldsymbol{w}} \right\|_2^2 = 2 H \frac{\partial \mathcal{L}}{\partial \boldsymbol{w}} \tag{56}$$

Substitute back and we have

$$\partial_{\boldsymbol{v}} \left\| \frac{\partial \mathcal{L}}{\partial \boldsymbol{w}} \right\|_2^2 = \boldsymbol{v} \cdot \left( 2 H \frac{\partial \mathcal{L}}{\partial \boldsymbol{w}} \right) \tag{57}$$

Since $\partial_{\boldsymbol{v}} \left\| \frac{\partial \mathcal{L}}{\partial \boldsymbol{w}} \right\|_2^2 = 0$ for all vector $\boldsymbol{v}$ that is orthogonal to $\frac{\partial \mathcal{L}}{\partial \boldsymbol{w}}$, $\boldsymbol{v} \cdot \left( 2 H \frac{\partial \mathcal{L}}{\partial \boldsymbol{w}} \right) = 0$ for all vector $\boldsymbol{v}$ that is orthogonal to $\frac{\partial \mathcal{L}}{\partial \boldsymbol{w}}$. In other words, $2 H \frac{\partial \mathcal{L}}{\partial \boldsymbol{w}}$ is orthogonal to all vectors that are orthogonal to $\frac{\partial \mathcal{L}}{\partial \boldsymbol{w}}$. Therefore, $2 H \frac{\partial \mathcal{L}}{\partial \boldsymbol{w}}$ has the same direction of $\frac{\partial \mathcal{L}}{\partial \boldsymbol{w}}$, and $\frac{\partial \mathcal{L}}{\partial \boldsymbol{w}}$ is an eigenvector of the Hessian of $\mathcal{L}$. $\square$

Proposition 5.7 is a direct consequence of Lemma C.4.

**Proposition 5.7.** *Let* $S_{\mathcal{L}_0} = \{\boldsymbol{w} : \mathcal{L}(\boldsymbol{w}) = \mathcal{L}_0\}$ *be a level set of* $\mathcal{L}$*. If at a particular* $\boldsymbol{w} \in S_{\mathcal{L}_0}$ *we have* $\|\nabla \mathcal{L}(\boldsymbol{w})\|_2 \geq \|\nabla \mathcal{L}(\boldsymbol{w}')\|_2$ *for all* $\boldsymbol{w}'$ *in a small neighborhood of* $\boldsymbol{w}$ *in* $S_{\mathcal{L}_0}$*, then the gradient* $\nabla \mathcal{L}(\boldsymbol{w})$ *has the same direction as the Newton's direction* $H^{-1} \nabla \mathcal{L}(\boldsymbol{w})$*.*

*Proof.* From Lemma C.4, $\frac{dL}{dw}$ is an eigenvector of $H$. Therefore, it is also an eigenvector of $H^{-1}$. Hence $\frac{dL}{dw}$ has the same direction as $H^{-1} \frac{dL}{dw}$. $\square$

# D Experiment details and additional results

## D.1 Test functions

We compare the gradient at different loss values for gradient descent with and without teleportation. Figure 7 shows that the trajectory with teleportation has a larger $d\mathcal{L}/dt$ value than the trajectory without teleportation at the same loss values. Therefore, the rate of change in the loss is larger in the trajectory with teleportation, which makes it favorable.

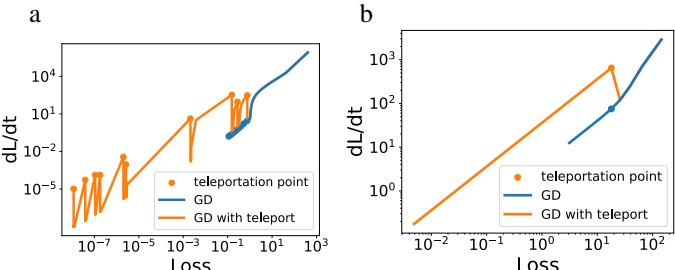

Figure 7: Gradient on the trajectory of optimizing the Rosenbrock function (left) and Booth function (right). At the same loss value, the graident is larger on the trajectory with teleportation, indicating a better descent path.

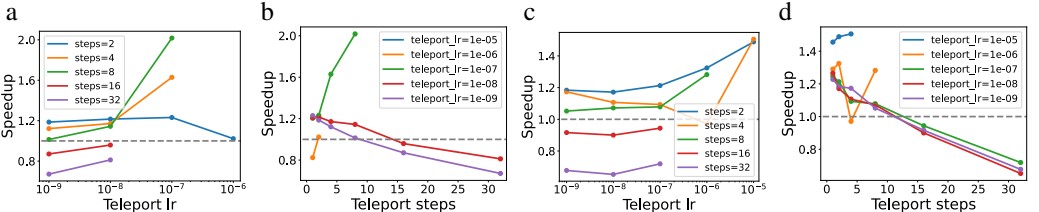

Figure 8: Hyperparameter sweeps of the number of steps and the learning rate used to find the optimal group element in teleportation. The wall-clock speedup of applying teleportation is shown separately for gradient descent (a)(b) and AdaGrad (c)(d). The dashed line represents speedup $= 1$.

## D.2 Multilayer neural network

**Additional training details** Data $X, Y$ and initialization of parameters $W$ are set uniformly at random over $[0, 1]$. GD uses learning rate $10^{-4}$ and AdaGrad uses $10^{-1}$. Each algorithm is run 300 steps. When using teleportation, we perform symmetry transform on the parameters once at epoch 5. In GD, the group elements used for these transforms are found by gradient ascent on $T$ for 8 steps, with learning rate $10^{-7}$. In AdaGrad, the group elements are found by gradient ascent for 2 steps, with learning rate $10^{-5}$. The choice of hyperparameters comes from a grid search described in the next section.

**Hyperparameter tuning** To observe the effect of hyperparameters on the speedup in computation time, we did a hyperparameter sweep on the number of steps and the learning rate used in each teleportation. The speedup of teleportation on SGD and AdaGrad is defined by $t_{sgd}/t_{sgd+teleport}$ and $t_{adagrad}/t_{adagrad+teleport}$ respectively, where $t_{sgd}, t_{adagrad}$ are the wall-clock time required to reach convergence using SGD or AdaGrad, and $t_{sgd+teleport}, t_{adagrad+teleport}$ are convergence time with teleportation. We consider the optimization algorithm converged if the difference between the loss of two consecutive steps is less than $10^{-3}$. This experiment is run on one CPU.

Figure 8 shows the speedup of the same multilayer neural network regression problem defined in Section 6.1, but teleporting only once at epoch 5. We did a grid search for teleportation learning rates in $[10^{-9}, 10^{-8}, 10^{-7}, 10^{-6}, 10^{-5}]$ and number of teleportation steps in $[1, 2, 4, 8, 16, 32]$. Omitted points in the figure indicate that the gradient descent fails to converge within 2000 steps or diverges.

When converged, most hyperparameter combinations improve the convergence speed in wall-clock time (speedup $> 1$). There are trade-offs in both the number of steps used for each teleportation and the teleportation learning rate. Increasing the number of steps used to optimize teleportation target allows us to find a better point in the parameter space but increases the cost of one teleportation. Increasing the learning rate of optimizing the group element improves $\|\partial\mathcal{L}/\partial\boldsymbol{w}\|$ but is more likely to lead to divergence since $\|\partial\mathcal{L}/\partial\boldsymbol{w}\|$ can become too large for the gradient descent learning rate.