# OpenReview forum: "Symmetry Teleportation for Accelerated Optimization"
_NeurIPS.cc/2022/Conference — NeurIPS 2022 Accept_

### Official Review · Reviewer_94ST · 2022-07-11

**Rating:** 6
**Confidence:** 2
**Soundness:** 3 good
**Presentation:** 4 excellent
**Contribution:** 3 good

**Summary:**

This work proposes an accelerated gradient-based optimization algorithm, symmetry teleportation, which exploits symmetries in the loss landscape. It improves current neural teleportation methods by searching for teleportation destinies that lead to the largest improvement in the magnitude of gradient. This work also provides empirical evidence showing the improved convergence speed of gradient descent and AdaGrad in different optimization problems.

**Questions:**

How to exploit symmetries for different loss functions？

**Limitations:**

There are no potential negative societal impacts of the work.

**Strengths And Weaknesses:**

Strength:
1. This work proposes a novel optimization algorithm that applies symmetric teleportation to accelerate the convergence of gradient descent.
2. The problem is well-motivated, and the work also provides an intuitive illustration of the ideas behind the proposed algorithm.

Weakness:
1. It would be better if the work could explain more about how to exploit symmetries for different loss functions.

---

> ### Author Response · Authors · 2022-08-02
> **Response to Reviewer 94ST**
>
> Thank you for your comments and positive feedback!
>
> > How to exploit symmetries for different loss functions?
>
> Since the symmetry comes from the architecture, we find the symmetries by observing what transformations of the parameters leave the loss function unchanged. For example, as shown in the paper, when there are two consecutive layers with elementwise activation functions, the neural networks admits a GL(R) symmetry. Another example is neural network that uses radial activation functions, where SO(n) is a symmetry. Kunin et al. (2021) also mention examples of symmetries in different neural networks include translational, scaling, and rescaling symmetry. Once we find the symmetry in the neural networks, we can apply teleportation by optimizing on the group element to improve convergence rate.

---

> > ### Comment · Reviewer_94ST · 2022-08-08
> > **Re:**
> >
> > Thank you for the clarifications!

---

### Official Review · Reviewer_ag4f · 2022-07-11

**Rating:** 6
**Confidence:** 3
**Soundness:** 2 fair
**Presentation:** 3 good
**Contribution:** 3 good

**Summary:**

Existing gradient-based optimization methods update the parameters locally, in a direction that minimizes the loss function. The author proposes a different optimization method to improve the convergence speed by teleportation which transforms parameters while making the loss invariant. The author gives theoretical proof and experimentally shows that teleportation improves the convergence speed of GD and AdaGrad for several optimization problems.

**Questions:**

1. The meaning of ‘Acceleration’ should be stated clearly in this paper.
2. Does the initialization have an impact on the convergence and the final result through teleportation? How and why?
3. The proliferation of saddle points can slow down convergence. Can Symmetry Teleportation effectively escape saddle points?
4. Is there any general guideline for designing symmetry groups?
5. In some scenarios, GD with a learning rate schedule can also achieve a similar loss curve in Fig. 2(c). What advantages do you think GD with teleport has over GD with a learning rate schedule?
6. As far as I can see, GD in deterministic settings is slightly different from GD in stochastic settings. How do you guarantee that SGD+teleport can accelerate convergence over SGD?
7. There is a little abuse of notations. The learning rate, $\eta$, and the learning rate matrix, $\eta$. The steps, T, and the transformation matrix, T.

Typos

Line 73 leads -> lead

Line 88 flow -> flows

Line 130 are -> is

Line 227 reduces -> reduce

Line 272 has -> have

Line 285 has -> have

Line 315 minima -> minimum

etc.

**Limitations:**

Yes

**Strengths And Weaknesses:**

Strengths:
1. The teleportation improves the convergence speed of subsequent steps by transforming parameters to another point with steeper gradients.
2. The symmetry teleportation takes advantage of higher-order landscape geometry but uses only gradient information.

Weaknesses
1. It does not guarantee that the transformed parameters lead to a faster convergence rate throughout the entire training.
2. It takes time to find g.
3. When the number of weights and data is large, it may increase the complexity.

---

> ### Author Response · Authors · 2022-08-02
> **Response to Reviewer ag4f (Part 2)**
>
> > 6. As far as I can see, GD in deterministic settings is slightly different from GD in stochastic settings. How do you guarantee that SGD+teleport can accelerate convergence over SGD?
>
> Whether teleportation accelerates the convergence of SGD depends on the data we use.
> The expected change in loss and convergence is related to the variance in the data as well as minibatch sizes. In our experiments, we observe that even teleporting using a small number of data points (e.g. 80 images in MNIST) is able to improve the convergence for the objective function trained on the entire dataset. This suggests that the loss landscape created by samples of data is similar to the landscape created using all available data.
>
>
> > 7. There is a little abuse of notations.
>
> Thanks for pointing these issues out. When using constant learning rates, we assume that $\eta$ is a scalar multiple of the identity matrix. We have added clarifications in the paper and changed the notation for the number of steps to $t_{max}$.

---

> > ### Comment · Reviewer_ag4f · 2022-08-09
> > **Response to authors**
> >
> > Thanks for addressing many questions in my original review. I have carefully read your responses. You have added a theoretical analysis of a class of optimization problems (quadratic convex functions), which is solid. That is great. Thanks for your efforts.
> >
> > We agree that the expected change in loss and convergence is related to the variance in the data as well as minibatch sizes. Empirical results have demonstrated that teleporting using a small number of data points (e. g.80 images in MNIST) is able to improve the convergence. It would be better if there could be a theoretical analysis, especially on the loss landscape.

---

> > > ### Author Response · Authors · 2022-08-09
> > > **Thank you and additional results**
> > >
> > > Thanks for reading our responses and proofs! We really appreciate it.
> > >
> > > We have updated the paper, where we added a new section Appendix C.5. We have formalized the improvement brought by teleportation in SGD and added some discussion about the gap between teleporting using mini-batches vs. all data. Due to time constraint, we are not able to finish the proof that gives a high probability bound. However, we would be happy to add more formal results in the final version.

---

> > > > ### Comment · Reviewer_ag4f · 2022-08-09
> > > > **Thanks for the response**
> > > >
> > > > I would like to thank the authors for the clarification. Now, it makes sense to me and I have raised the score.

---

> ### Author Response · Authors · 2022-08-02
> **Response to Reviewer ag4f (Part 1)**
>
> Thank you for the insightful comments and detailed suggestions! We have made the suggested edits in the paper and address the main questions below.
>
> **Response to weaknesses**
> > 1. It does not guarantee that the transformed parameters lead to a faster convergence rate throughout the entire training.
>
> In the new Appendix D, we show that for a class of optimization problems (quadratic convex functions) teleporting once guarantees optimality at all future times.
> For general problems, we can do multiple teleportations.
> Thus, to observe faster convergence rate each teleportation only needs to increase $|d\mathcal{L}/dt|$ until next teleportation.
> We also provide a condition for when one teleportation gives the optimal trajectory. We consider a trajectory optimal if for every point on the trajectory, the magnitude of gradient is at a local maximum in the loss level set that contains the point.
>
>
> > 2. It takes time to find g.
> > 3. When the number of weights and data is large, it may increase the complexity.
>
> Yes, the computational complexity of teleportation depends on the dimension of weights and the number of samples in a mini-batch. As shown in section 6.3, one teleportation step has the same complexity as back propagation (albeit with a different constant). Since we only teleport a few times during training, the additional time brought by teleportation is small compared to the speedup in convergence.
>
>
> **Response to questions**
> > 1. The meaning of ‘Acceleration’ should be stated clearly in this paper.
>
> By “acceleration” we mean increasing $|d\mathcal{L}(g \cdot \mathbf{w})/dt|= \|\nabla \mathcal{L}(g\cdot \mathbf{w})\|^2 $ the gradient norm and decreasing the number of epochs required to converge. We have added clarifications in the introduction. We also added empirical results of wall-clock convergence time in Figure 4d, 9, and 10.
>
> > 2. Does the initialization have an impact on the convergence and the final result through teleportation? How and why?
>
> Yes, but teleportation reduces the effect of initialization on the convergence rate because it moves parameters in the landscape to a point with better gradient norm. If all points on a level set are reachable by a group action, and we teleport the parameters to the optimal point right after initialization, then initialization does not have an impact on the convergence. In practice, however, the group actions are usually not transitive, and we find the optimal teleportation destination by gradient ascent on a non convex function. Therefore, part of the effect of initialization remains.
>
>
> > 3. The proliferation of saddle points can slow down convergence. Can Symmetry Teleportation effectively escape saddle points?
>
> That is an interesting point. Yes, teleportation can be an effective technique to escape saddle points. At saddle points, $d\mathcal{L}(g \cdot \mathbf{w})/dt = 0$. By definition, teleportation moves parameters in the directions to increase $d\mathcal{L}(g \cdot \mathbf{w})/dt$.   Additionally, since teleportation can be applied at any time during training, we can teleport every time when we notice a decrease in $d\mathcal{L}(g \cdot \mathbf{w})/dt$, which may help us move away from saddle points.
>
>
> > 4. Is there any general guideline for designing symmetry groups?
>
> Yes, by observing the form of the optimization function. The choice of symmetry group depends entirely on the architecture. For example, as shown in the paper, when there are two consecutive layers with elementwise activation functions, the neural networks admits a GL(R) symmetry. Another example of continuous symmetry is a neural network that uses radial activation functions, where SO(n) is a symmetry.
>
>
> > 5. In some scenarios, GD with a learning rate schedule can also achieve a similar loss curve in Fig. 2(c). What advantages do you think GD with teleport has over GD with a learning rate schedule?
>
> Teleportation is an  orthogonal technique to  learning rate schedule. We can use teleportation in conjunction with learning rate schedule. In fact, since we have some idea of how the magnitude of gradient changes with teleportation (increases then decreases quickly), teleportation may benefit significantly from a learning rate schedule.

---

### Official Review · Reviewer_W8uy · 2022-07-11

**Rating:** 6
**Confidence:** 4
**Soundness:** 3 good
**Presentation:** 3 good
**Contribution:** 3 good

**Summary:**

Using teleportation of parameters to a new point at the same loss value but with a larger gradient norm, the authors offer a new approach for accelerating the convergence of gradient-based optimization methods. This new scheme can accelerate the convergence of gradient-based optimization methods. An intuitive understanding of the connection between such a scheme and second-order methods is demonstrated by the authors. The major application discussed in the paper are deep neural networks and two rather simple classical functions. An empirical evaluation of the effectiveness of the authors' approach to solving these problems was provided, along with a comparison to gradient descent and AdaGrad.

**Questions:**

1) There are two main assumptions for unconstrained optimization problems: $L$-smoothness (Lipschitz-continuous gradient) and $M$-Lipschitz-continuous function. Both are defined globally but we are actually using them locally on the set around the starting point $x_0$ and the solution $x_{\ast}$. The reason why we can do it is the monotonicity and compactness of gradient-based methods. With that local approach, we can make our function class much broader. For example, x4 is not $L$-smooth or $M$-Lipschitz, but we can constrain $L$ locally by some constant dependent on $x_0$, and the gradient method will converge for such a $L$. Note that the presented Rosenbrock function is not $L$-smooth or $M$-Lipschitz globally. The same problem arises for neural networks. Additionally, level sets can be unbounded. This leads us to the question of the existence of such teleportation because the maximization problem will also be unbounded. For example, one of the simplest examples of neural networks is a linear 2-layer network without activations for one data-point $(1,1)$ with MSE-loss.  $\min (x \cdot y - 1)^2$. All points $x = 1/y$ are local minima. Let us start from parameters $a = (2,1) $. The gradient norm at $b = (1000, 0.002)$ is bigger than the gradient norm of $a$. So, we'll teleport to $b$, but why is this location superior? Furthermore, because of smoothness, the learning rate at $b$ should be less than $1/(1000^2+0.002^2)<10^{-6}$, whereas the learning rate at $a$ is around $1/5$. So, my main concern is that this teleportation technique can actually make convergence worse and much slower. Because of that, the method should be used much more carefully and with clear limitations to its applicability.

2) It is unclear to me, how complex the process of teleportation is in general.

3) Some of the notations and objects were unclear to me. So, I would recommend adding a notation section to the appendix to clarify the introduced notation. For example, what does $g \cdot w$ mean? Why are you using $dL/dt$ and not $dL/dw$? Some definitions of used groups or links to the literature where the reader can be educated about them will also be helpful.

4) In experiments,
a) Figure 4: Why are SGD and AdaGrad starting from different points?
b) How do you tune learning rates for gradient descent? Do
c) Figure 5: It seems that the fact that teleportation has bad validation accuracy proves my claim from question 1.
d) Figure 3 and 8: It seems that the gradients of the teleported version of AdaGrad are smaller than regular AdaGrad. Despite this, the teleported loss is smaller than regular. How does it correspond with the dependence "bigger gradient -> faster convergence"?

5) The authors showed intuition about two-dimensional quadratic problems (ellipsoid). It will be interesting to understand what will happen with d-dimensional quadratic problems.

#######################
During the rebuttal, the authors addressed most of my concerns and questions. They improved the paper. Special thanks for the new Section D.1 about quadratic problems. I increased my overall score to 6.

**Limitations:**

The authors adequately addressed the limitations.

**Strengths And Weaknesses:**

The symmetry teleportation technique is the main idea and contribution of the paper. It is based on the fact that after teleportation, the loss will be the same, but the landscape can be much better, and hence we will get faster convergence to the local minima. I liked the presented method's relations with second-order methods both theoretically and practically on Booth function. The idea is interesting and original. Some proofs are presented, and they seem to be correct. The authors show some experiments for small feed-forward networks with leak-relu activations on MNIST. The paper is well-structured and well-written. I have some concerns about the correctness and effectiveness of the teleportation technique presented in the paper. I will address them in the questions section.

---

> ### Author Response · Authors · 2022-08-02
> **Response to Reviewer W8uy (Part 2)**
>
>
>  **Q4:** questions about experiments
>
> > a) Figure 4: Why are SGD and AdaGrad starting from different points?
>
> We show the loss after each epoch, so the loss after the first epoch appears different for SGD and AdaGrad.
> Note that we only compare SGD with SGD-teleport (not with AdaGrad), and compare AdaGrad with AdaGrad-teleport.
> Therefore, SGD and AdaGrad having different initial loss doesn't affect our results.
>
> We noticed that teleportation was applied at different times for SGD and AdaGrad on MNIST. For consistency, we re-ran the experiment with teleportation in the same epoch and updated Figure 4.
>
> > b) How do you tune learning rates for gradient descent?
>
> We performed hyper-parameter tuning using the validation set. We choose the learning rates such that further increases affect the converged value.
> In our experiments, gradient descent with teleportation does not require a different learning rate from regular gradient descent.
> Teleportation improves convergence even when gradient descent uses different learning rates, as shown in Figure 3 and 8.
>
> > c) Figure 5: It seems that the fact that teleportation has bad validation accuracy proves my claim from question 1.
>
> Since teleportation is able to improve the train accuracy, we believe that the bad validation accuracy is not caused by the unbounded maximisation problem or inappropriate learning rates. As we noted in the experiment section,
> a possible explanation would be a sharp local minima, which have large gradients but may generalize poorly.
> We suspect using different or larger minibatches of data to do teleportation might alleviate this or reveal if sharp local minima are indeed the culprit.
>
>
> > d) Figure 3 and 8: It seems that the gradients of the teleported version of AdaGrad are smaller than regular AdaGrad. Despite this, the teleported loss is smaller than regular. How does it correspond with the dependence "bigger gradient -> faster convergence"?
>
> Actually, the interpretation of the plots is opposite to this.
> Let us clarify.
> In Figure 3b and 8b, at the same epoch , the gradients of the teleported versions are smaller.
> However, this can be explained by the faster convergence of the teleported versions, and that the magnitude of gradients is typically smaller when we are closer to the minima. This is proven by the fact that in Figure 3c and 8c, at the same loss values, the teleported versions have a larger $d\mathcal{L}(g \cdot \mathbf{w})/dt$, meaning that teleportation helped find a better trajectory.
>
>
> > **Q5:** It will be interesting to understand what will happen with d-dimensional quadratic problems.
>
>
> Great suggestion!
> Indeed we have exact results (Appendix D.1) proving that a single teleportation can find the fastest descending trajectory for convex quadratic loss in $n$ dimensions.
>
> In brief,
> consider a quadratic form $L_A(w) = \frac{1}{2} w^T A w$, where $w \in \mathbb{R}^n$ is the parameter and $A \in \mathbb{R}^{n \times n}$ is a diagonal matrix with positive diagonal elements. Then the level sets of $L_A$ are $n$-dimensional ellipsoids centered at 0, with axes in the same direction as the standard basis.
>
> The gradient of $L_A$ is $\nabla L_A = Aw$, and the magnitude of the gradient is $\|\nabla L\|^2 = \|Aw\|$. The point with largest $\|\nabla L\|^2$ on a level set is in the eigendirection of $A$ corresponding to its largest eigenvalue, or the point on the smallest semi-axes.
> The gradient flow trajectory from this point always points to the global minima at 0. Therefore, like the 2D ellipse function, one teleportation on the $n$-dimensional ellipsoid also guarantees optimal convergence rate at all points along the trajectory.
>
> This analysis can be made more general by considering any positive definite matrix $A$. A more formal statement and additional details can be founded in the new section D.1 in the appendix of the revised paper.

---

> ### Author Response · Authors · 2022-08-02
> **Response to Reviewer W8uy (Part 1)**
>
> Thank you for your comments! We appreciate the insights on local Lipschitz and smoothness constraints. We address the main questions below.
>
> > **Q1:** This leads us to the question of the existence of such teleportation, because the maximisation problem will also be unbounded. ... my main concern is that this teleportation technique can actually make convergence worse and much slower. Because of that, the method should be used much more carefully and with clear limitations to its applicability.
>
> Thank you for pointing this out.
> We agree that the method should be applied with care, but since we don't modify the loss landscape in any way, we still believe the method is applicable in general to any optimization problem having symmetries.
> Teleportation is simply like initializing at a point with steeper gradients.
> You are correct that
> the optimal group element can be unbounded, and teleportation can lead to divergence if we do not bound the parameters during optimization.
> We propose two approaches to address this problem.
> First, we can stop the optimization process when the magnitude of gradient is larger than a pre-set threshold. Second, we can use adaptive learning rates.
> For example, we can set a lower learning rate immediately after teleportation and increase to its original value after a few steps.
>
>
> > **Q2:** how complex the process of teleportation in general
>
> Teleportation is a simple gradient descent when the symmetry is known.
> It is as easy as linear regression for linear symmetries and for an MLP with nonlinear activations, it only involves an extra function that implements the group action.
> Also, note that we do not have to teleport all layers (e.g. we can only teleport the last two fully-connected layers in VGGNet).
> As we showed in sec. 4.2, even neural networks with nonlinear activations can have continuous symmetries.
> Deriving the teleportation process only requires finding the transformation of parameters that leaves the loss unchanged (e.g. Proposition 4.3).
> In our paper, the data is not transformed, so transforming parameters may involve finding the pseudoinverse of the output of specific layers, which is quadratic in the number of rows of layer output and linear in the minibatch size. As an example, the runtime complexity for a multi-layer neural network is given in section 6.3.
>
>
> > **Q3:** Some of the notations and objects were unclear
>
> $g \cdot w$ denotes the group action of a group element $g$ on parameters $w$. We refer to $|d\mathcal{L}(g \cdot \mathbf{w})/dt|$ as the "convergence rate", which is equal to the $L_2$ norm of $\partial \mathcal{L}(g \cdot \mathbf{w})/\partial \mathbf{w}$ in gradient flow. We have added a reference to basic group theory in the appendix.

---

### Official Review · Reviewer_WGhM · 2022-07-16

**Rating:** 6
**Confidence:** 4
**Soundness:** 4 excellent
**Presentation:** 4 excellent
**Contribution:** 2 fair

**Summary:**

Authors consider the class of optimization problems having group of symmetry G, which is simple enough to optimize norm of the gradient over its orbits, and propose technique of "teleportation" from current point to the more suitable one on the same orbit before every gradient step. They present the examples of problems having symmetries, including simplest NN, and numerically show the competitiveness of proposed technique in these simple cases and more complicated test problems (MNIST). New technique is provided with sketchy theoretical justification of acceleration effect.

**Questions:**

Regarding second weakness — it would be good to find at least one case (class of functions + class of symmetries) when GD with teleportation is competitive with GD/Conjugate gradients method/Newton method with explicitly expressed effect of teleportation in final convergence rate. It seems to be possible.

**Limitations:**

Limitations seem to be obvious for the readers of the paper, and authors do not pretend to propose the panacea.

**Strengths And Weaknesses:**

The idea of using group theory in optimization for solving problems with known symmetries is not new (I've found review with DOI:10.1007/978-1-4614-1927-3_9 almost immediately, but it is hardly the earliest), but this topic was not well discovered for continuous convex case. The first strength of the paper is that it attracts attention to this blank space. The reason why it is blank is the unnaturalness of group of symmetry oracle — symmetries that cannot be used to simplify the model itself are often too complicated to be used in auxiliary problems as well. This is related with first weakness — paper does not consider question of automatic symmetry detection and does not say anything about practical NN architectures, but this is very important to show that this technique at least can be applied to any practical task. The second weakness is that sketch of theoretical analysis is too sketchy, there are no explicit convergence rate estimations in the paper, so the effect of teleportation cannot be appreciated.

---

> ### Author Response · Authors · 2022-08-02
> **Response to Reviewer WGhM**
>
> Thank you for your comments and insight on this field.
>
> > paper does not consider question of automatic symmetry detection and does not say anything about practical NN architectures, but this is very important to show that this technique at least can be applied to any practical task.
>
> Note that Section 4.2 addresses this question for multi-layer perceptrons (fully-connected layers) in NNs.
> The key point is that for two consecutive layers with element-wise activation within any NN, no symmetry detection is needed. For defining the symmetry transformations in Proposition 4.3 one only needs
> the model architecture.
> We believe Proposition 4.3 can be generalized to other NN layer types (e.g. CNN or attention layers).
> In neural networks, both the architecture and the loss function are known, so finding parameter-space symmetry does not require automatic symmetry detection which is often used in discovering symmetry in data.
>
>
> > there are no explicit convergence rate estimations in the paper, so the effect of teleportation cannot be appreciated.
>
> For convex quadratic functions, we have added a result that one teleportation is guaranteed to improve convergence rate along the entire trajectory in Appendix D.1.
> We do not have explicit bounds for global convergence rate, which is difficult to obtain for non-convex non-smooth problems. However, we did introduce conditions of when $d\mathcal{L}/dt$ can be improved in the subsequent steps after a teleportation (Section 5.2).
> We also provided intuitions of the convergence rate by relating teleportation to second order methods, and showed empirical evidence in test functions and MNIST classification.
>
>
> > it would be good to find at least one case (class of functions + class of symmetries) when GD with teleportation is competitive with GD/Conjugate gradients method/Newton method with explicitly expressed effect of teleportation in final convergence rate.
>
> In the new Appendix D, we added a class of functions where teleporting once guarantees optimality at all future times. For general functions, we also provide a condition for when one teleportation gives the optimal trajectory. We consider a trajectory optimal if for every point on the trajectory, the magnitude of gradient is at a local maximum in the loss level set that contains the point.

---

### Official Review · Reviewer_ysBc · 2022-07-17

**Rating:** 7
**Confidence:** 3
**Soundness:** 4 excellent
**Presentation:** 4 excellent
**Contribution:** 3 good

**Summary:**

This paper proposes moving along the level set during an optimization problem. Doing so may lead to faster gradient iterations, and is proven and shown empirically. A few examples of symmetries are discussed (modifications to the parameters that leave the output of the network invariant). For a nonlinear MLP, only an input-dependent symmetry is discussed. Runtime analysis is also performed.

**Questions:**

 * Does the theory behind convergence take into account the extra steps needed to solve for the symmetry teleportation problem? I'm wondering if the symmetry teleportation problem itself has a different convergence rate, and the real convergence rate would be worse?

 * Practically, how would Figure a look if it is plotted against number of gradient computations or even wall clock time? The runtime analysis per teleportation is great to have, but since this cost of teleportation is amortized (done only every so GD iterations), it isn't completely clear to me how this actually affects practice.

 * Since only a first order approximation is used for exp(x), the output of the neural network isn't invariant anymore right? Moreover, if you take more teleportation steps, this should actually diverge or at least cause the parameters to be arbitrarily far away from the level set. What's the intuition behind this not occurring in practice; is it due to the fixed number of steps for optimizing the teleportation?

Minor:

 * How did you settle on 10 steps for teleportation? How important is it to tune this number?

 * Given your expertise, would you use symmetry teleportation as a default when you run optimization problems with known symmetries? Why or why not.

 * Small suggestion for Figure 2 b and f: draw the symmetry teleportation trajectory. Right now some of the blue lines are gradient descent while others are teleportation + GD which makes it difficult to parse the details.

 * Type (line 62): dt -> dw

**Limitations:**

.

**Strengths And Weaknesses:**

Pros:
  - Paper is well-written.
  - Problem setting (optimization of MLPs) is relevant.
  - Proposed solution is simple to understand.

Cons:
  - Runtime complexity is not fully reconciled / transparent.
  - Some details of the symmetry teleportation is not completely clear to me.
  - In practice, symmetries must be known for a particular optimization problem, and even when known, in many cases I imagine optimizing within this group can be very difficult.

---

> ### Author Response · Authors · 2022-08-02
> **Response to Reviewer ysBc (Part 1)**
>
> Thank you for your comments!
>
> **Response to the cons and main questions**
>
> > Runtime complexity is not fully reconciled / transparent.
>
> > Practically, how would Figure a look if it is plotted against number of gradient computations or even wall clock time? The runtime analysis per teleportation is great to have, but since this cost of teleportation is amortized (done only every so GD iterations), it isn't completely clear to me how this actually affects practice.
>
> Thanks for the suggestion! We added Figures 3(d) and 4(d) that show loss vs. wall clock time for MLP. On randomly distributed data, teleportation slows down optimization, which is expected since the overhead is large for computing group actions using the entire training data. On MNIST, teleportation has negligible effect on training time since only a few mini-batches are used. The speedup with respect to wall clock time is therefore similar to the speedup observed with respect to the number of iterations.
>
> > In practice, symmetries must be known for a particular optimization problem, and even when known, in many cases I imagine optimizing within this group can be very difficult.
>
> While obtaining the complete set of symmetries of an arbitrary loss function is hard, finding a subset of the symmetries is often easy. For example, if there are two consecutive layers with element-wise activation, which is common in neural networks, we are able to teleport using the symmetry created by this structure alone.
> Regarding optimization within the symmetry group $G$, our method is tractable for the following reasons:
> * We do not seek a globally optimal $g\in G$, instead $g$ only needs to improve $d\mathcal{L}/dt$ until the next teleportation.
> * Since we work with continuous symmetry, small changes in the group element can be approximated to first order as $g \approx I+\varepsilon T$, where $T\in \mathfrak{g}$ is in the Lie algebra of $G$. This approximation avoids the expensive computation of the exponential map.
> * The constraints on $T$ are generally easy to satisfy. For example, when $G=GL_d(\mathbb{R})$, $T$ can be any arbitrary $d\times d$ matrix.
> Thus, for $GL_d(\mathbb{R})$ we can simply do gradient descent on the unconstrained optimization $\max_g d\mathcal{L}(g\cdot \mathbf{w})/dt$.
>
>
> > Does the theory behind convergence take into account the extra steps needed to solve for the symmetry teleportation problem? I'm wondering if the symmetry teleportation problem itself has a different convergence rate, and the real convergence rate would be worse?
>
> No, the convergence rate of the optimization is decoupled with computational cost of teleportation steps in our analysis. In theory, teleportation improves the rate but may be more computational costly. However, since we do not teleport at every step, the cost of teleportation is amortized. We also show empirically that teleportation reduces the overall computation time in the new Figure 10.
>
>
> > Since only a first order approximation is used for exp(x), the output of the neural network isn't invariant anymore right? Moreover, if you take more teleportation steps, this should actually diverge or at least cause the parameters to be arbitrarily far away from the level set. What's the intuition behind this not occurring in practice; is it due to the fixed number of steps for optimizing the teleportation?
>
> Up to $O(\varepsilon^2)$, the loss remains invariant.
> When making $k$ teleportation steps, as long as the $O(\varepsilon^2)$ term is still negligible, the approximation error is not significant.
> We show here that $O(\varepsilon^2)$ terms are negligible when $k\varepsilon \ll 1$.
>
> The Taylor expansion of the exponential map is $g=\exp[\varepsilon T]= I+\varepsilon T +O(\varepsilon^2)$. When applying $k$ steps of teleportation with $g_i = I+\varepsilon T_i$ we have a total $g$ given by
> $$g = \prod_{i=1}^k g_i = I + \varepsilon \sum_{i=1}^k T_i + \varepsilon^2 \sum_{i,j=1}^k T_iT_j + O(\varepsilon^3) $$
>
> The error introduced by the first-order approximation is $O(\varepsilon^2)$, which is small since $\varepsilon \ll 1$.
> The number of $O(\varepsilon)$ terms is $k$ (one for each $T_i$) and number of $O(\varepsilon^2)$ terms is $k(k-1)/2$ ($k$ choose $2$ of $T_i$).
> Therefore, as long as $(k-1)\varepsilon/2 \ll 1$, the higher order terms can be ignored.

---

> ### Author Response · Authors · 2022-08-02
> **Response to Reviewer ysBc (Part 2)**
>
> **Response to the minor questions**
>
> > How did you settle on 10 steps for teleportation? How important is it to tune this number?
>
> We hand tuned this number. The number of steps, together with the learning rate used to optimize the group element $g$, are chosen such that $d\mathcal{L}(g \cdot \mathbf{w})/dt$ shows clear improvement but does not become large enough to cause divergence.
> When $d\mathcal{L}(g \cdot \mathbf{w})/dt$ is not bounded on a loss level set such as in multi-layer neural networks, it is important to restrict the steps for teleportation to a finite value, although in practice the effect of teleportation is not sensitive to the number of steps.
>
> We added a hyperparameter sweep to study the effect of hyperparameters on the speedup in computational time. In the new Figure 9, most hyperparameter combinations improve the convergence speed in wall-clock time.
>
> > Given your expertise, would you use symmetry teleportation as a default when you run optimization problems with known symmetries? Why or why not.
>
> Yes, we would, though there are trade-offs to consider.
> Symmetry teleportation has a clear benefit of improving convergence rate and reducing training cost.
> However, the computational overhead of implementing teleportation and hyperparameter tuning may limit its advantages.
> Further investigation is needed to find the best settings to apply teleportation.
>
> > Small suggestion for Figure 2 b and f: draw the symmetry teleportation trajectory. Right now some of the blue lines are gradient descent while others are teleportation + GD which makes it difficult to parse the details.
>
> Thanks for the suggestion! We have updated that figure to make the teleportations clearer.
>
> > (line 62): dt $\to $ dw
>
> We are rewording this sentence.
> We aimed to say: "In comparison, we search within $G$-orbits for points which maximize $|d\mathcal{L}(g\cdot \mathbf{w})/dt| = \varepsilon^2 \|\nabla \mathcal{L} (g\cdot\mathbf{w})\|^2 $."

---

### Author Response · Authors · 2022-08-02
**Revision summary**


We thank the reviewers for their insightful comments. We are encouraged that they find our proposed teleportation method novel, intuitive, and well-motivated. We are also glad that Reviewer W8uy likes the connection between our algorithm and second-order methods.
We have added individual response for each reviewer and updated the paper.
We would also like to summarize the comments shared among all reviewers and major revisions in the new version of the paper.

* **Detecting symmetries in general optimization problems:**
Detecting symmetries is not the focus of this paper.
Nevertheless, for neural networks, we did discover a novel set of symmetries (Proposition 4.3) for fully-connected layers.
For general optimization problems, we need to know the symmetries to use teleportation. This is usually not hard since we have access to the optimization function.

* **Wall-clock time:**
We have added wall-clock time and hyperparameter sweep in Appendix E.2.
There we show that teleportation can indeed improve wall-clock time for convergence, in addition to reducing the number of epochs.
Also, wall-clock time can be optimized further via more efficient implementation of the teleportation steps.

* **Guaranteed speedup:**
We have added a new Appendix D, in which we show that for quadratic convex optimization one teleportation is enough to guarantee faster convergence. We then develop a general condition for when one teleportation gives the optimal trajectory.

---

### Author Response · Authors · 2022-08-08
**Appreciate your feedback on our responses**

Dear reviewers and AC,

We have submitted our reviews and comments for our submission. We would really appreciate it if you could give us feedbacks on the response, or at least the rebuttal acknowledgement. We hope to have an open and scientific conversation about our work!

Thank you for your time!

---

### Meta-Review · Area_Chair_k2EX · 2022-08-26

**Recommendation:** Accept
**Confidence:** Certain

**Metareview:**

This paper proposes a novel, symmetry teleportation approach to optimize the parameters of ML models.
The proposed approach allows iterates to move along the loss level set and improves the convergence speed. The teleportations also exploit the symmetries that are present in the optimization problem.
The paper also includes very encouraging numerical experiments.
I believe that the paper brings more insides and techniques that have been mostly overlooked in the community when training ML problems.

**Award:**

No

---

### Decision · Program_Chairs · 2022-09-14

Accept